# Early-life exercise primes the murine neural epigenome to facilitate gene expression and hippocampal memory consolidation

Anthony M. Raus [1,6], Tyson D. Fuller[2,6], Nellie E. Nelson [1,6], David A. Valientes[2], Anita Bayat[2] & Autumn S. Ivy [1,2,3,4,5] ✉

Aerobic exercise is well known to promote neuroplasticity and hippocampal memory. In the developing brain, early-life exercise (ELE) can lead to persistent improvements in hippocampal function, yet molecular mechanisms underlying this phenomenon have not been fully explored. In this study, transgenic mice harboring the "NuTRAP" (Nuclear tagging and Translating Ribosome Affinity Purification) cassette in Emx1 expressing neurons ("Emx1-NuTRAP" mice) undergo ELE during adolescence. We then simultaneously isolate and sequence translating mRNA and nuclear chromatin from single hippocampal homogenates containing Emx1-expressing neurons. This approach allowed us to couple translatomic with epigenomic sequencing data to evaluate the influence of histone modifications H4K8ac and H3K27me3 on translating mRNA after ELE. A subset of ELE mice underwent a hippocampal learning task to determine the gene expression and epigenetic underpinnings of ELE's contribution to improved hippocampal memory performance. From this experiment, we discover gene expression – histone modification relationships that may play a critical role in facilitated memory after ELE. Our data reveal candidate gene-histone modification interactions and implicate gene regulatory pathways involved in ELE's impact on hippocampal memory.

[1] Physiology/Biophysics, Anatomy/Neurobiology, University of California- Irvine School of Medicine, Irvine, CA, USA. [2] Pediatrics, University of California-Irvine School of Medicine, Irvine, CA, USA. [3] Neurobiology/Behavior, University of California- Irvine School of Biological Sciences, Irvine, CA, USA. [4] Anatomy/Neurobiology, University of California- Irvine School of Medicine, Irvine, CA, USA. [5] Division of Neurology, Children's Hospital Orange County, Orange, CA, USA. [6] These authors contributed equally: Anthony M. Raus, Tyson D. Fuller, Nellie E. Nelson. ✉email: aivy@hs.uci.edu

Environmental experiences engage epigenetic mechanisms to modulate gene expression and cell function in post-mitotic neurons[1,2]. Histone modifications and DNA methylation are particularly important for neuronal adaptation to environmental signals by altering transcription and synaptic function[3,4]. Behavioral outputs such as stress susceptibility, reward seeking, and long-term memory have been shown to involve changes to chromatin accessibility and gene expression in neurons[5–9]. In addition to this, the neuronal chromatin landscape undergoes waves of epigenetic modifications as a function of brain maturation itself[10–12]. Postnatal periods of heightened sensitivity to environmental stimuli can lead to lasting changes to cellular function and may result from temporally specific epigenetic mechanisms in the developing brain[13,14]. Whether gene regulatory mechanisms in postmitotic neurons are uniquely influenced by early-life experiences to inform long-term function is a question that is just beginning to be explored. Identifying epigenetic processes involved in modulating neuron function, particularly during brain development, is critical for understanding how early-life experiences impact long-term behavioral outcomes.

Aerobic exercise enhances performance on cognitive tasks involving the hippocampus in both adult humans and animal models[15,16]. The type, timing, and duration of exercise exposure matters with regard to whether it has a persistent impact on hippocampal function[17–19]. Findings in both adolescent and adult rodents implicate a role for histone modifying enzymes in the mechanisms of exercise-induced benefits to hippocampal memory. Both voluntary exercise or treatment with a HDAC3 inhibitor enable hippocampal memory after a subthreshold learning stimulus, increase brain-derived neurotrophic factor (BDNF) gene expression, and promote acetylation of H4K8 at the promoters of specific BDNF exons[20–22]. This suggests that exercise engages epigenetic regulatory mechanisms to promote hippocampal plasticity. Interestingly, adult exercise also opens a temporal window for persistent improvements to hippocampal memory performance when a reactivating exercise exposure is introduced[23], suggesting a "molecular memory" of the initial exercise[23]. Although many of these studies have been performed in adults, more recent studies demonstrate that the effects of exercise on hippocampal memory, alterations in neurotrophic factor expression, synaptic plasticity, and neurogenesis are similar in juvenile and adolescent periods[17,24–27]. Previous findings from our lab demonstrate that early-life exercise (ELE) for either one week (juvenile period; postnatal days (P) 21–27) or three weeks (juvenile-adolescence; P21–41) facilitate hippocampal long-term memory formation in response to a learning stimulus typically insufficient for forming long-term memory. This finding was associated with increased long-term potentiation (LTP) as well as modulations to synaptic physiology in hippocampal CA1[17]. Notably, the hippocampal memory effects of juvenile ELE persisted two weeks after exercise cessation, which is potentially longer than the effect of exercise on adult hippocampal function[23]. Taking these findings together, it is possible that exercise (whether in early life or adulthood) may "prime" hippocampal function for facilitated responses to future experiences (such as future exercise bouts or hippocampal learning events). Epigenetic mechanisms are strong candidates for the priming effects of exercise in that the epigenome could represent a molecular memory of the exercise experience by readying the chromatin landscape for efficient gene expression, thereby modulating neuronal function and behavioral output[22]. The specific mechanisms underlying sustained behavioral and electrophysiological effects of ELE have not been assessed from the perspective of a potential molecular memory of ELE within the epigenome.

In this study we simultaneously isolate neuron-specific chromatin and translating mRNA[28] from ELE mice to identify (and directly correlate) gene expression programs with governing histone modifications engaged by ELE to improve hippocampal memory and plasticity. By crossing an Emx1-Cre transgenic line with mice harboring the nuclear tagging and translating ribosome affinity purification (NuTRAP)[29] cassette (termed "Emx1-NuTRAP" mice), we isolate whole nuclei and translating mRNA from a single hippocampus by combining the first steps of the isolation of nuclei tagged in specific cell types (INTACT[30]) and translating ribosome affinity purification (TRAP[31]) protocols. We call this technical approach simultaneous INTACT & TRAP (SIT) and have published methodological details elsewhere[28]. This technique yields DNA and RNA from the same homogenate that is of high-quality and sufficient concentrations for downstream sequencing procedures (in this study, TRAP-seq and CUT&RUN-seq). Using neuron-specific, paired transcriptomic and epigenomic sequencing, these data reveal a possible "epigenetic signature" resulting from ELE that could contribute to alterations in gene expression underlying improved hippocampal memory.

## Results

**"Emx1-NuTRAP" mouse allows for simultaneous isolation of nuclear chromatin and translating mRNA from a single population of hippocampal neurons**. Our initial goal was to obtain translating mRNA and nuclear DNA from a single population of neurons to couple gene expression changes with alterations in histone modifications resulting from ELE. To do this, we developed a transgenic mouse line (Emx1-NuTRAP) and experimental protocol (SIT protocol[28]). We crossed male NuTRAP reporter mice[29] with female Emx1-Cre[32,33] mouse lines to generate Emx1-Cre; NuTRAP transgenic progeny for this study ("Emx1-NuTRAP"; Fig. 1a). Emx1-expressing neurons are predominately excitatory[32] and are necessary for hippocampal neurogenesis and motor skill learning[34]. In the presence of a *loxP* site-flanked sequence, Emx1-Cre facilitates recombination in approximately 88% of neurons in the neocortex and hippocampus, and in less than 2% of GABAergic inhibitory interneurons in these regions[32]. Emx1-Cre also targets neural progenitor cells and can be expressed in mature astrocytes in striatum[35]. Of note, we exclusively used Emx1-Cre female mice in our breeding schemes given that the Cre-recombinase has been reported to be expressed in male germline cells[36].

We first sought to validate neuron-specific expression of the NuTRAP cassette. Immunohistochemichal analysis of Emx1-NuTRAP hippocampal slices show distinct GFP and mCherry expression in CA1 and CA3 pyramidal neurons (Fig. 1b, c) as well as granule cells of the dentate gyrus (DG; Fig. 1d). As Emx1-expressing neural stem cells of the DG can potentially differentiate into astrocytes[32], we sought to determine whether a significant number of mature astrocytes were also obtained in the population of isolated cells. Flow cytometry experiments using Thy1 as a neuronal marker and S100β as an astrocytic marker revealed a distinct population of cells double positive for GFP and Thy1, whereas a S100β and GFP double positive cell population was absent (Fig. 1e; see Supplementary Fig. 1, and Supplementary Data 1 for gating). There was a small population of GFP+/Thy1− singlets. We interpret this finding to reflect Emx1 labeling of dividing immature neural progenitors, as Thy1 is not expressed in granule cells of the DG until the cells are 4-5 weeks old[37].

To perform simultaneous INTACT & TRAP ("SIT")[28], hippocampal tissue was dissected from both brain hemispheres, combined, and homogenized in one sample tube (Fig. 1a). We

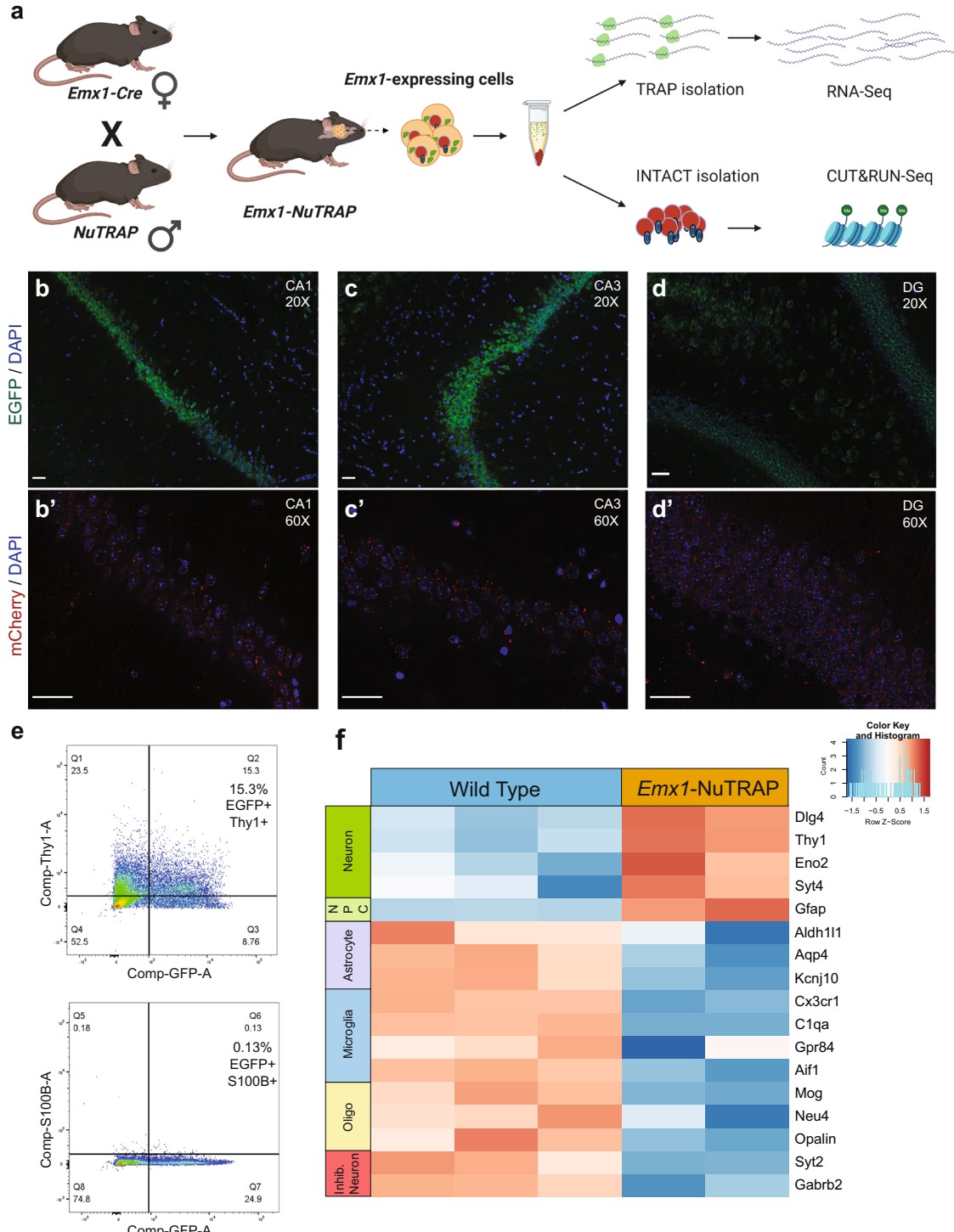

**Fig. 1 Generation and validation of the Emx1-NuTRAP mouse as a tool for generating sequencing-grade, neuron-specific DNA and translating mRNA from a single hippocampal homogenate. a** Schematic representation of the Emx1-NuTRAP mouse generation and workflow of the Simultaneous INTACT and TRAP ("SIT") protocol. Created with BioRender.com. **b** Immunofluorescence imaging of the CA1 region of the hippocampus at ×20 objective and **b'** ×60 objective after incubation with an mCherry antibody . **c** Immunofluorescence imaging of the CA3 region of the hippocampus at ×20 objective and **c'** ×60 objective after incubation with an mCherry antibody . **d** Immunofluorescence imaging of the DG region of the hippocampus at ×20 objective and **d'** ×60 objective after incubation with an mCherry antibody . Scale bars are set to 50 μm for all images **b–d**. **e** Flow cytometry for EGFP, Thy1, and S100β (15.3% of cells Thy1 + /EGFP + and 0.13% of cells S100β+/EGFP+). **f** Heatmap of differentially expressed neuronal and non-neuronal cell type markers from RNA-seq data comparing TRAP-isolated RNA from hippocampus of Emx1-NuTRAP mice vs hippocampal mRNA isolated from wild-type mice. (Abbreviations: NPC neural progenitor cell, Oligo oligodendrocyte).

then performed the SIT method on isolated samples by starting the protocol with the beginning steps from the INTACT procedure modified to include cycloheximide. Cycloheximide works rapidly to inhibit protein synthesis and is used for maintaining crosslinks between translating mRNA and ribosomal subunits during purification in the traditional TRAP method. Despite the presence of cycloheximide, nuclear morphology from Emx1-expressing neurons was unchanged, as demonstrated by mCherry positive nuclei from hippocampal neurons remaining successfully bound to magnetic beads after our modified INTACT procedure (Supplementary Fig. 2a). Additionally, a Bioanalyzer was used to determine if there were cycloheximide-induced double-stranded DNA breaks in the nuclear preparations that could potentially interfere with downstream DNA sequencing applications (such as ATAC-, CUT&RUN-, or CUT&Tag-seq). We found no evidence of DNA double strand breaks generating fragments of <1 kb[28]. Following magnetic purification of biotin-labeled nuclei, the supernatant fraction was removed and taken through TRAP, while the pelleted nuclei were processed through the remaining steps of INTACT (see Methods section and ref. [28]). Using qPCR, we found that TRAP-isolated mRNA had significantly reduced expression of *Mog* and *Cd11b* compared to total RNA, suggesting depletion of oligodendrocyte and microglial populations in TRAP mRNA, respectively (Supplementary Fig. 2b). Combining bilateral hippocampi from a single mouse yielded TRAP-isolated mRNA of high quality and sufficient concentration for sequencing (RIN > 8 for all samples, average yield RNA = 14.367 ng/ul; Supplementary Data 2)[28]. INTACT-isolated nuclei were further processed using the cleavage under targets and release using nuclease (CUT&RUN[38]) method to isolate antibody-specific protein-DNA interactions for downstream DNA sequencing. The resulting DNA libraries were of high quality and concentration when using specific antibodies (H4K8ac: average size = 1238 bp, average concentration = 126.8 nM; H3K27me3: average size = 1032 bp, average concentration = 146.5 nM; Supplementary Data 2). In contrast, the resulting DNA libraries using the non-specific IgG control had substantially lower concentrations (IgG: average size = 1035 bp, average concentration = 24.2 nM; Supplementary Data 2) further indicating that nuclear DNA from both isolations was of high starting quality.

To confirm our TRAP-isolated hippocampal mRNA came primarily from excitatory neurons, we compared RNA-seq data from hippocampal tissue of wild type mice vs TRAP-isolated RNA-seq (TRAP-seq) data from Emx1-NuTRAP mice to assess for neuronal gene enrichment. We found that TRAP-isolated mRNA had enrichment of several neuronal genes, including *Dlg4*, *Thy1*, *Eno2*, and *Syt4*, as well as a substantial reduction in astrocytic, microglial, oligodendrocyte, and inhibitory neuronal genes (Fig. 1f; for a heat map with expanded gene list taken from[39], see Supplementary Fig. 2c). There was a relative expression increase of glial fibrillary astrocytic protein (GFAP) in our TRAP samples. GFAP can also be expressed in neural stem populations that were present in our whole hippocampus samples, so this finding may reflect the neural stem cell population of the dentate gyrus[40,41]. Overall, these findings support the Emx1-NuTRAP mouse model as a valid tool for neuron-enriched isolation of sequencing-grade, translating mRNA and nuclear chromatin from a single brain tissue homogenate.

**ELE reveals highly comparable translatomic and epigenomic profiles resulting from simultaneous and separate DNA and RNA isolation methods.** Previous methods for isolating DNA and mRNA using INTACT and TRAP (respectively) from NuTRAP mouse tissue have taken separate tissue homogenates for each procedure[29,42]. In this study, we developed an approach to perform SIT on a single tissue homogenate containing bilateral hippocampi[28] (the products of SIT are herein referred to as "simultaneous isolations"). To determine if our SIT approach yields comparable results to INTACT and TRAP performed on separate samples, we isolate hippocampal tissue obtained from a separate brain hemisphere for each method and counterbalanced for left versus right (we refer to this protocol as "separate isolations"). A Bioanalyzer was used to determine the amount and quality of the RNA obtained from each type of isolation (Separate isolations: average RNA concentration = 7.805 ng/ul, RIN > 8 for all samples; Simultaneous isolations: average RNA concentration = 14.367 ng/ul, RIN > 8 for all samples; Supplementary Data 2). The average RNA yield from the separate isolations (using a unilateral hippocampus) was approximately equal to half of the average yield of the simultaneous isolations (which combined bilateral hippocampi; Supplementary Data 2). Similarly, the final library concentrations for the separately isolated IgG CUT&RUN-seq libraries were also approximately half the concentration of the simultaneous isolations (average simultaneous: 24.2 nM, average separate: 11 nM; Supplementary Data 2). We interpret this to mean that nuclear DNA was fully intact in the simultaneous isolation because we did not obtain substantially more than double the concentration in the simultaneous vs separate isolations. Unilateral hippocampal homogenates yielded sufficient sequencing concentrations and quality to allow for library preparations from individual mice (Supplementary Data 2).

To determine if normalized TRAP-seq counts were similar between the two isolation methods, we performed a Spearman's correlation between the datasets from sedentary mice. The two TRAP-seq datasets were found to be highly correlated, with R values >0.5 and p value < $2.2 \times 10^{-16}$ (R = 1, p < $2.2 \times 10^{-16}$; Fig. 2a). Next, CUT&RUN-seq was used to identify genomic regions interacting with two histone post-translational modifications (PTMs) of interest in the exercise and hippocampal memory fields: H4K8ac, a permissive histone mark associated with active transcription, or H3K27me3, a generally repressive histone PTM. We again applied Spearman's correlation to understand whether the simultaneous vs separate INTACT isolation methods could influence CUT&RUN-seq peak distribution. We compared normalized count data for CUT&RUN-seq peaks across a representative chromosome (chromosome 2). We binned 100 bp increments along the entire chromosome from the simultaneous and separate isolations using datasets generated from sedentary mice in our study. Normalized sequencing counts, which reflected reads assigned to binned genomic regions along chromosome 2, were highly similar between separate vs simultaneous conditions (H4K8ac: R = 0.65, p < $2.2 \times 10^{-16}$; H3K27me3: R = 0.81, p < $2.2 \times 10^{-16}$; Fig. 2b, c). Taken together, these experiments suggest that translating mRNA and nuclear DNA isolated from hippocampal homogenates using either simultaneous or separate isolation procedures are comparable in terms of quality, concentration, functional characterization, and normalized sequencing reads.

We next wanted to determine if the different isolation methods could bias resulting gene expression on the basis of gene length. The differentially expressed genes (DEGs) were analyzed using DESeq2 and Student's t-tests to determine statistically significant differences in RNA-seq and detect potential biologically relevant log-fold changes. When plotting gene length against log fold change of gene expression, we see a similar distribution pattern of ELE-induced hippocampal DEGs between the simultaneous and separate methods (Fig. 2d, e). We then compared the biological categories of the ELE-induced DEGs obtained using either the separate or the simultaneous isolation methods using Panther Gene Ontology (GO). Although many of the genes did not

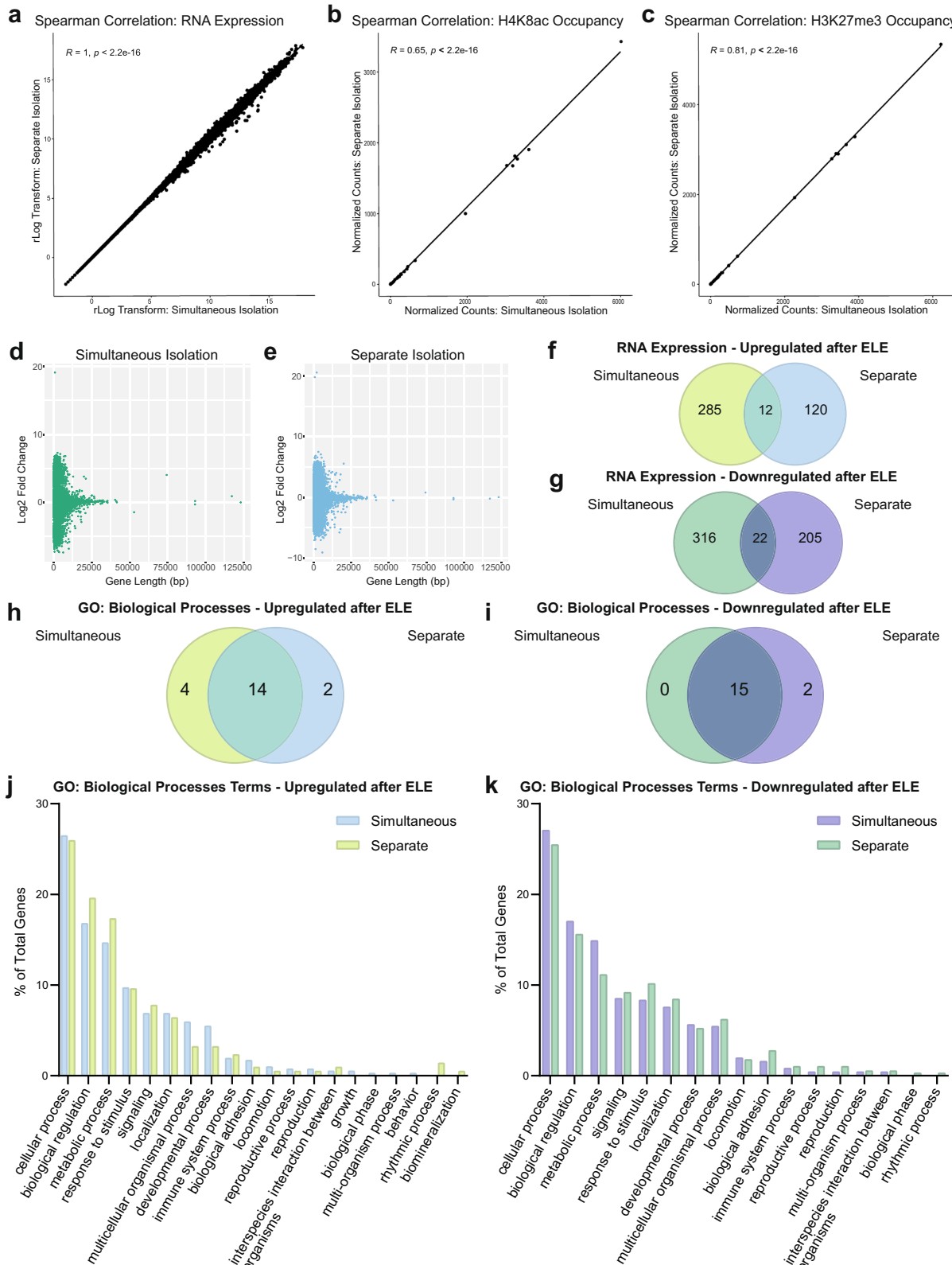

overlap (Fig. 2f, g and Supplementary Data 3), most GO terms did (14/20 GO terms for upregulated genes and 15/17 for downregulated genes; Fig. 2h, i and Supplementary Data 3). Furthermore, there was high similarity between the percentage of genes found in each of the GO categories (Fig. 2j, k). These results demonstrate that the simultaneous approach for isolating and sequencing mRNA and nuclear chromatin can be performed

using a single hippocampal homogenate and can generate comparable results to more traditional methods using separate cell samples to pair different types of sequencing data.

**Neuronal gene expression programs resulting from ELE, and histone PTM peak distribution, are biologically comparable**

**Fig. 2 Comparing simultaneous (SIT) and separate DNA and RNA isolation procedures from Emx1-NuTRAP mice that underwent ELE.** For **a–e**, $n = 2$ mice for simultaneous isolations and n = 3 mice for separate isolations. **a** Spearman's correlation for RNA expression between separately isolated RNA and simultaneously isolated RNA ($R = 1$, $p < 2.2 \times 10^{-16}$). **b** Spearman's correlation between separate and simultaneously isolated DNA from CUT&RUN-seq for H4K8ac with 100 bp chromosome 2 position bins ($R = 0.65$, $p < 2.2 \times 10^{-16}$). **c** Spearman's correlation between separate and simultaneously isolated DNA from CUT&RUN-seq for H3K27me3 with 100 bp chromosome 2 position bins ($R = 0.81$, $p < 2.2 \times 10^{-16}$). **d** Gene length versus fold change distribution plot for simultaneously isolated RNA. **e** Gene length versus fold change distribution plot for separately isolated RNA. **f** Genes upregulated in TRAP-seq after ELE in either simultaneous ($n = 2$ sedentary mice and $n = 3$ ELE mice) or separate isolations ($n = 3$ mice per group). **g** Genes downregulated in TRAP-seq after ELE in either simultaneous or separate isolations. **h**, **i** Panther Gene Ontology: Biological Processes Venn diagram for genes upregulated (**h**) or downregulated (**i**) after ELE in either separate or simultaneous isolations. **j**, **k** Panther Gene Ontology: Biological Processes bar graph of gene distributions for genes upregulated (**j**) and downregulated (**k**) after ELE in either separate or simultaneous isolations.

**between left and right hippocampal hemispheres**. Many studies take advantage of the brain's structural symmetry by using tissue from each hemisphere for separate molecular processing. Prior evidence demonstrates that hippocampal lateralization can influence LTP, hemisphere-specific glutamate receptor density, and performance in certain memory tasks[43]. We wanted to determine if there were differences in transcriptional programs and biological processes resulting from ELE in left vs right hippocampi. Using the separate isolation approach described above, we compared ELE-induced differential gene expression between left and right hemispheres. We found that although individual gene expression patterns were different between hippocampi originating in the left and right brain hemispheres (Fig. 3a, b and Supplementary Data 4 and 5), the Panther GO: Biological Processes terms were similar, with most categories overlapping (Fig. 3c–f and Supplementary Data 4). Furthermore, applying 2-way ANOVA and likelihood ratio statistical analyses with multiple comparisons find more genes associated with the "exercise" term than "hemisphere" or "interaction" (2-way ANOVA: "exercise" = 1583, "hemisphere" = 808, "interaction" = 869; Likelihood Ratio Test: "exercise" = 565, "hemisphere" = 466, "interaction" = 548; Supplementary Data 5). suggesting a greater impact of ELE over hemisphere on differential gene expression, and the possibility of spatial or hemispheric assignment of genes with functional similarity.

We next determined if left and right hippocampal hemispheres had significant differences in DEGs and CUT&RUN-seq peak distributions at baseline (without exercise). We performed Spearman's correlation on the transcript counts from the TRAP-seq data ($R = 1$, $p < 2.2 \times 10^{-16}$; Fig. 3g), and the CUT&RUN-seq 100 bp binned peak counts along a representative chromosome (chromosome 2) for two histone modifications: H4K8ac ($R = 0.59$, $p < 2.2 \times 10^{-16}$; Fig. 3h), and H3K27me3 ($R = 0.70$, $p < 2.2 \times 10^{-16}$; Fig. 3i). We found that both the transcript counts and CUT&RUN-seq peaks were highly similar (significance considered as $R$ values >0.5 and $p$ value < $2.2 \times 10^{-16}$; Fig. 3g–i). Overall, left and right hippocampal hemispheres did not demonstrate significant differences in normalized sequencing counts and peak distributions in the sedentary condition, suggesting that choice of hippocampal hemisphere is a less important factor to consider in obtaining representative data from the translatome and transcriptional regulation via histone modifications.

**ELE promotes expression of plasticity-related genes and implicates transcriptional regulatory pathways involved in hippocampal memory.** To identify the gene expression programs induced by ELE in hippocampal neurons, we performed neuron-enriched bulk sequencing on TRAP-isolated mRNA extracted from Emx1-NuTRAP mouse hippocampi using our "SIT" protocol. Using DESeq2 to assess for DEGs (>30% expression increase with a $p$ value < 0.05) our TRAP-seq revealed 297 upregulated and 338 downregulated hippocampal genes resulting from ELE (Fig. 4a and Supplementary Data 3). Many of the genes

upregulated after ELE are known to be involved in exercise and/or hippocampal memory mechanisms, including *Bdnf*[44] and *Nr4a1*[45]. To functionally categorize ELE-induced DEGs, we performed a Panther Gene Ontology (GO) analysis[46] focusing on the Molecular Function categorization and separated by upregulated and downregulated genes. Regardless of gene expression directionality, GO term categories with the most genes functionally assigned to them were "binding", "catalytic activity", "molecular function regulator", "transporter activity", "molecular transducer activity", and "structural molecule activity" (Fig. 4b). Many of the upregulated genes driving these categories are known to have critical roles in neuronal function (*Kcna1, Slc24a4, Stxbp5l, Gabra2*, and *Camk2n2*), neurodevelopment (*Artn, Kdm7a, Sox21, Gap43*, and *Efna5*), and hippocampal memory (*Bdnf, Nr4a1* and *Dusp5*).

To evaluate possible transcription factors and upstream regulators implicated by ELE-induced activated gene networks, we applied Qiagen's Ingenuity Pathway Analysis (IPA) to our TRAP-seq dataset[47]. We identified significant canonical signaling pathways implicated in ELE effects on hippocampal neuronal function (Supplementary Data 6). The top six IPA-identified "upstream regulators" by significance included CREB1, KMT2A, LEF1, SMAD3, EGR1, and MITF (Fig. 4c and Supplementary Data 6)[48–55]. The transcription factors KMT2A and LEF1 have not been previously associated with the effects of exercise on hippocampal function. Interestingly, CREB1 (through its associated CBP[56]) and KMT2A (through its methyltransferase activity[49]) both have histone modifying properties. LEF1 has been shown to regulate neural precursor proliferation in the hippocampus[50]. SMAD3 is critical for intermediate progenitor cell survival and negatively regulates serum irisin after exercise[51,52]. EGR1 is an immediate early gene that recruits TET1 DNA demethylase during neural activation and development[53]. MITF was the sixth upstream regulator identified and is a transcription factor linked to autophagy mechanisms[54]. These specific transcription factors were not significantly differentially expressed in our TRAP-seq dataset; however, one possibility is that their activity may not be linked to a change in their own expression but rather modulated by exercise to influence downstream gene expression.

To further understand whether the transcriptional profiles resulting from ELE were enriched for specific a priori assigned molecular functions, we performed Gene Set Enrichment Analysis (GSEA)[57]. We evaluated the "Reactome" category of functional gene sets followed by a leading-edge analysis to further determine which genes were driving the significant categories (Fig. 4d and Supplementary Data 7). Of the DEGs upregulated after ELE, several interesting categorizations and the genes driving them were revealed. The genes *H3c4, H2bc3, H2bc5*, and *H2bc9* genes, which encode histone family member proteins, were driving the categories: "DNA methylation", "HDACs deacetylate histones", and "transcriptional regulation by smRNAs". "Meiosis" and "meiotic recombination" also emerged in these results which was

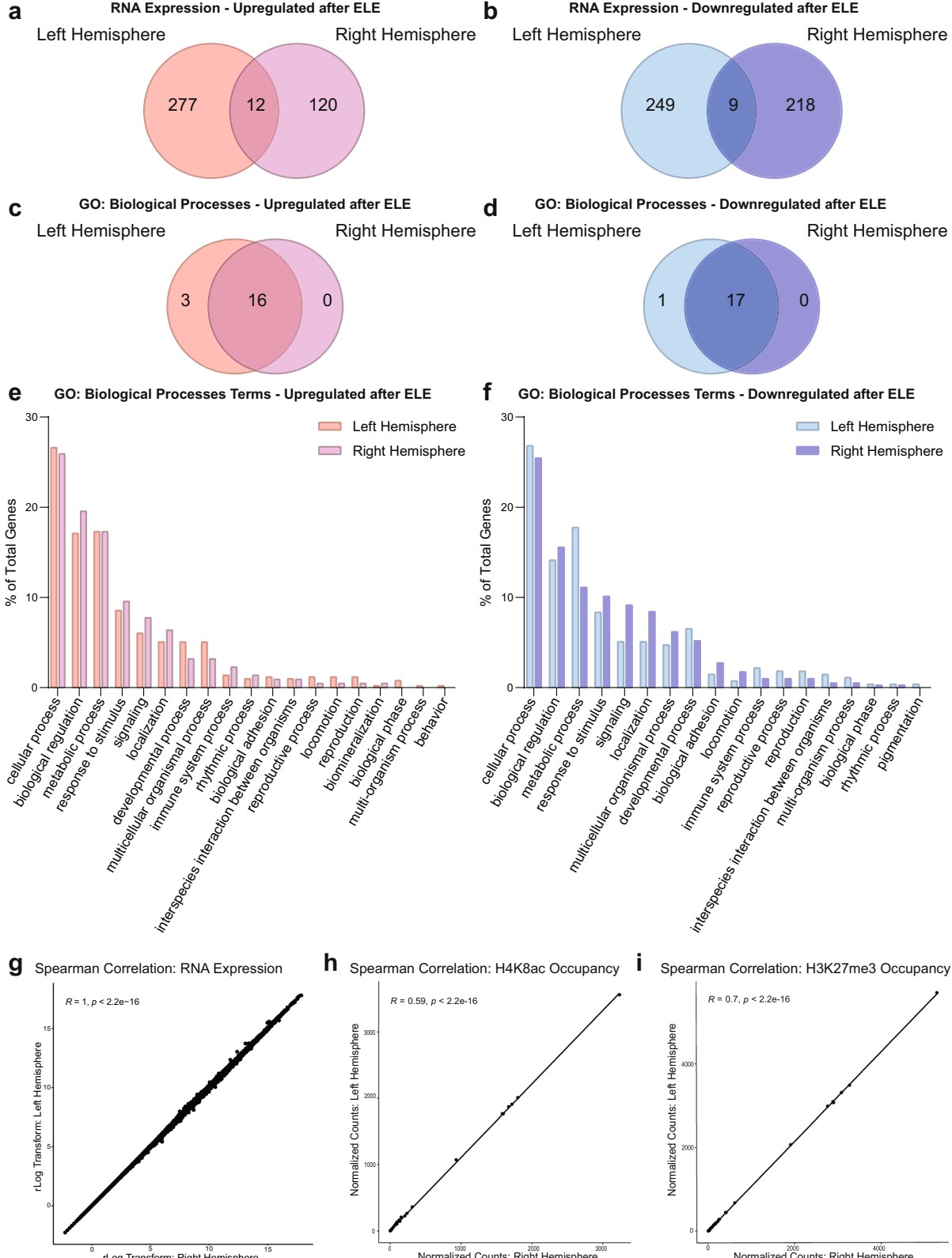

unusual; however, leading edge analysis showed many of these genes (*Atm, Blm, Msh4,* and *Mnd1*) to be generally involved in cell-cycling processes. Exercise is well known to increase adult neurogenesis in hippocampal dentate gyrus and can explain cell cycle gene enrichment in our dataset[16]. Several nucleoporin complex genes (*Nup35/ 42/ 43/ 50/ 58/ 62/ 93/ 107/ 153/ 210*) were also identified for their associations with categories such as:

"gene silencing by RNA", "nuclear envelope breakdown", and "transport of mature mRNAs derived from intronless transcripts". *Nup* genes form a variety of nuclear pore complexes that play critical roles in cellular processes including cell-cycle regulation, cellular differentiation, and epigenetic control[58].

Of the downregulated pathways identified, translation-associated categories ("translation", "eukaryotic translation

**Fig. 3 Functional categorizations of ELE-induced differential gene expression, and distribution of H4K8ac and H3K27me3, are similar regardless of hippocampal hemisphere. a, b** Venn diagram of genes upregulated (**a**) or downregulated (**b**) by ELE identified between the left and right hemispheres by the separate isolation protocol TRAP-seq (*n* = 3 mice per group). **c, d** Venn diagram of Panther Gene Ontology: Biological Process terms for genes upregulated (**c**) or downregulated (**d**) by ELE identified between the left and right hemispheres by the separate isolation protocol for performing TRAP-seq. **e-f** Percent of genes fitting into each GO category for the left and right hemisphere for genes upregulated (**e**) and downregulated (**f**) by ELE by separate isolation TRAP-seq. **g** Spearman's correlation between the left and right hemispheres TRAP-seq data from the separate isolation protocol of sedentary rlog normalized expression (*R* = 1, *p* < 2.2 × 10$^{-16}$). **h** Spearman's correlation between hemispheres for CUT&RUN-seq for H4K8ac normalized count data for chromosome 2 in 100 bp bins (*R* = 0.59, *p* < 2.2 × 10$^{-16}$). **i** Spearman's correlation between hemispheres for CUT&RUN-seq for H3K27me3 normalized count data for chromosome 2 in 100 bp bins (*R* = 0.7, *p* < 2.2 × 10$^{-16}$).

initiation/elongation", and "SRP-dependent co-translational protein targeting to membrane") were driven by significant downregulation of ribosomal protein gene families (*Rps*, *Rpl*, *Rplp*) and eukaryotic initiation factor 3 (*eIF3*) subunits. Over-expression of *eIF3* subunits has been linked to neurodegeneration and altered expression of *eIF3* has been associated with neurodevelopmental disorders[59]. Additionally, collagen genes (*Col1a1*, *Col1a2*, *Col2a1*, *Col3a1*, and *Col5a1*) were downregulated after ELE, leading to the identification of categories including: "MET activates PTK2 signaling", "MET promotes cell motility", and "collagen degradation". *Col1a1* and *Col1a2* have been previously identified as putative aging genes that decrease in their expression after chronic exercise in female adult rodents[60]. By analyzing these enriched pathways and the networks of genes driving them, we were able to identify several expected and unexpected transcriptional programs activated by ELE that could be unique to exercise timing during the juvenile-adolescent period. Ultimately, to determine if gene expression programs identified here are in fact unique to exercise timing, an adult exercised cohort would need to be included for comparison. This would be an important follow-up study to these data.

**Neuronal H4K8ac is enriched and H3K27me3 is reduced at a subset of plasticity genes after ELE.** The power of the Emx1-NuTRAP mouse model coupled with the "SIT" technical approach is the ability to directly pair differential gene expression with governing epigenetic mechanisms. This is achieved by obtaining both translatomic and epigenomic-sequencing data from the same cell population (Fig. 1a). To investigate ELE-induced changes in histone PTMs across the genome, hippocampal nuclei were obtained from ELE and sedentary mice using the INTACT method followed by CUT&RUN-seq for the modifications H3K27me3 and H4K8ac from nuclear chromatin. We chose these two histone PTMs for their previously described functions and potential roles in exercise and memory mechanisms. H4K8ac is a permissive modification that is enriched at *Bdnf* after adult exercise, and its presence correlates with improved hippocampal memory[22]. H3K27me3 is a repressive histone mark and a common control in CUT&RUN-seq studies[38,61] and is decreased at the *Bdnf* promoter region after contextual fear conditioning training[61]. Peaks were called using SEACR[62] and histone PTM enrichment peaks were overlapped with ELE-induced DEGs obtained from our TRAP-seq experiments described above (Fig. 5a and a', and Supplementary Fig. 3 for Upset plot). We evaluated for the presence of H4K8ac and H3K27me3 peaks in union with upregulated and downregulated mRNA resulting from ELE. 93 upregulated genes had new H4K8ac peaks, while 35 downregulated genes had new H3K27me3 peaks (Fig. 5b and Supplementary Data 8). Notably, of those 93 upregulated genes with H4K8ac peaks, 14 also had new H3K27me3 peaks resulting from ELE. 17 genes had presence of H3K27me3 in both conditions (ELE and sedentary; Fig. 5a). We interpret these results to mean that transcription of these 93 genes was promoted as a result of ELE-induced H4K8ac.

The loss of either H3K27me3 or H4K8ac after ELE did not directly correlate with directionality of translating mRNA expression changes (Fig. 5b). This result may indicate that ELE-induced differential gene expression is associated with the addition, rather than removal, of these chosen two histone modifications. We found one gene that was downregulated and had loss of H4K8ac (*Asphd1*), and two genes downregulated with loss of H3K27me3 (*Fdxr* and *Rtl1*; Fig. 5b and Supplementary Data 8). Of particular interest, reduced expression of paternally-inherited *Rtl1* is associated with improved hippocampal memory performance, whereas overexpression results in memory deficits[63,64]. Unsurprisingly, the majority of histone PTM peaks newly present as a result of ELE did not correlate with gene expression changes. However, notable upregulated genes with new H4K8ac after ELE include *Prox1*, *Sntb2*, and *Kptn*. PROX1 is involved governing differentiation, maturation, and DG versus CA3 cell identity in intermediate progenitor cells of the hippocampal DG[65], while SNTB2 is involved in G protein-coupled receptor cell signaling[66]. KPTN encodes a protein involved in cytoskeletal cell structure, and its mutation can cause neurodevelopmental disability and seizures[67]. Interestingly, downregulated genes found to have new H3K27me3 after ELE included *Col3a1*, *Efnb2*, *Epop*, and *Myoc*. COL3A1 and MYOC are structural proteins involved in the extracellular matrix and the cytoskeleton respectively[68–70]. EFNB2 is a signaling molecule involved in cell migration and its haploinsufficiency can cause neurodevelopmental disability[71–73]. EPOP is a known editor of the chromatin landscape by altering H2Bub and H3K4me3 distributions[74,75].

Although changes in gene expression associated with H3K27me3 peaks were less numerous than those associated with H4K8ac, we noticed a striking difference in the distribution of peaks across the genome (Fig. 5c). After ELE, the distribution of H3K27me3 decreases in the promoter region and increases in the distal intergenic and intronic regions (Fig. 5c). To investigate how ELE-induced changes to histone PTM distribution might alter gene expression, we looked at the distribution of peaks around a representative gene, *Prox1* (Fig. 5d). Most apparent changes in histone PTM peaks occurred in regions annotated as regulatory regions, rather than the promoter region, which is concordant with our data showing that most histone PTM changes happen outside of the promoter region. Taken together, these results suggest that ELE promotes the new addition of H4K8ac and H3K27me3 PTMs to regulate gene expression. Given that most identified peaks did not correlate with DEGs, they may instead implicate regions of interest that are "primed" for facilitated gene expression following future stimuli, such as learning.

**ELE alters H4K8ac and H3K27me3 occupancy at regulatory regions of genes implicated in hippocampal memory consolidation.** Prior work from our lab and others has found that both ELE and adult exercise can facilitate hippocampal long-term memory (LTM) formation in mice exposed to a typically sub-threshold learning event (3 min of object location memory

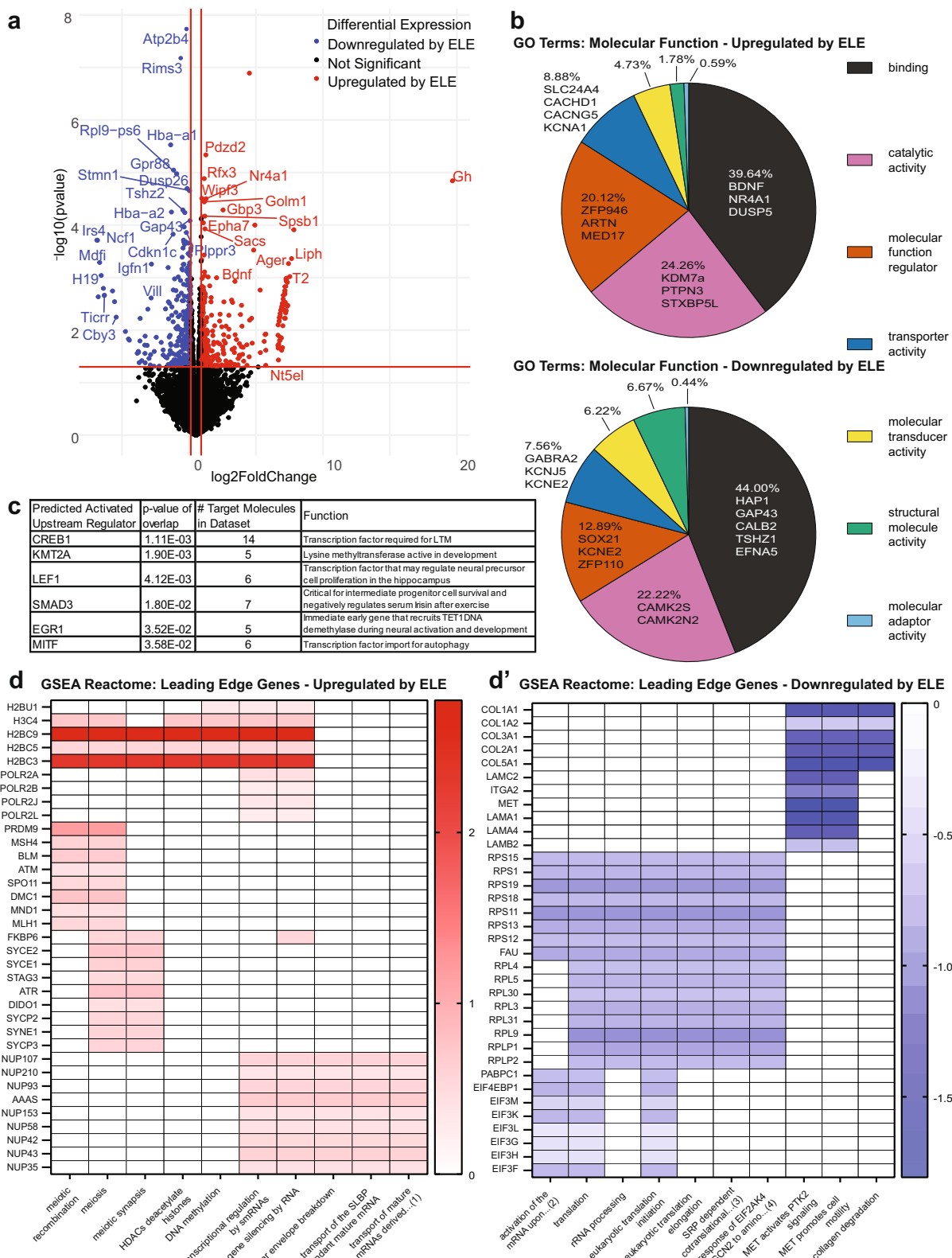

**Fig. 4 ELE results in novel transcriptomic changes in hippocampal neurons during adolescence. a** Volcano plot of differentially expressed genes reaching significance identified by TRAP-seq on ELE versus sedentary ($n = 2$ sedentary mice and $n = 3$ ELE mice, absolute value $\log_2$ fold change $>0.3785$ and $p$-value $< 0.05$). **b** Panther Gene Ontology: Molecular Function top terms by most genes assigned. **c** Top 6 "Upstream Regulators" identified by Ingenuity Pathway Analysis (IPA). **d** Representative Gene Set Enrichment Analysis: Reactome leading-edge diagrams showing genes upregulated (**d**) or downregulated (**d'**) in ELE, and their categories of enrichment. *Abbreviated terms in **d**: (1) "transport of mature mRNAs derived from intronless transcripts", **d'**: (2) "activation of the mRNA upon binding of the cap binding complex and EIFs and subsequent binding to 43S", (3) srp-dependent cotranslational protein targeting to membrane, and (4) response of eif2ak4 gcn2 to amino acid deficiency.

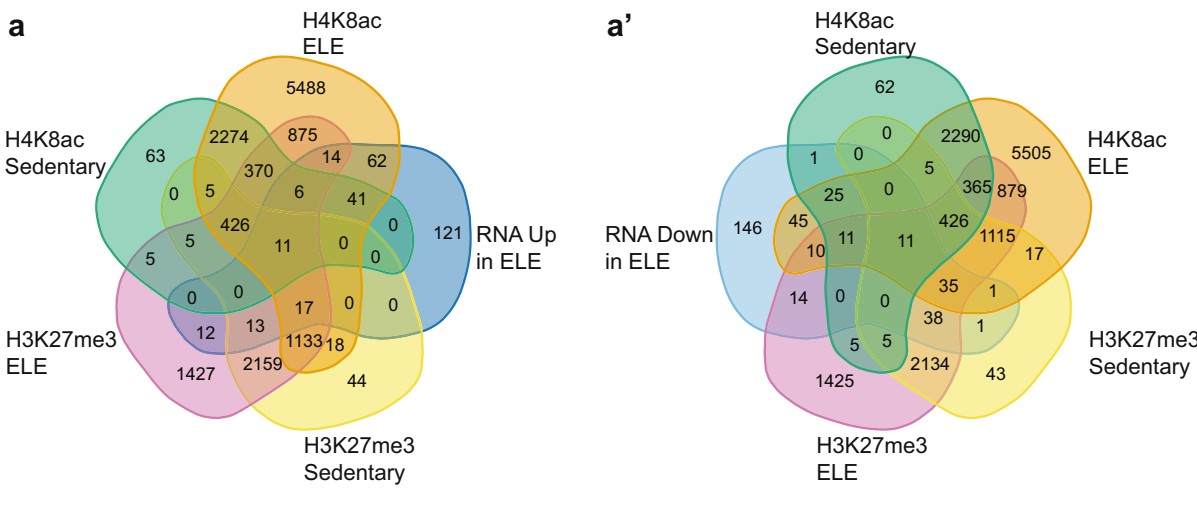

**Fig. 5 ELE alters H4K8ac and H3K27me3 occupancy at differentially expressed genes. a** Venn diagram of gene peaks identified in CUT&RUN-seq for H4K8ac and H3K27me3 by SEACR (top 1% of peaks FDR < 0.1) overlapped with genes upregulated (**a**) and downregulated (**a'**) in RNA by ELE (n = 2 sedentary mice and n = 3 ELE mice). **b** CUT&RUN-seq SEACR peak calls at genes that overlap with genes differentially expressed by ELE as determined by TRAP-seq. **c** Stacked bar of the relative distribution of peak calls at various genomic regions identified by CUT&RUN-seq called using SEACR. **d** Representative track of CUT&RUN-seq data around *Prox1* showing differential peak height between ELE and sedentary conditions for H4K8ac (top) and H3K27me3 (bottom).

(OLM) training, which is normally insufficient for LTM formation in sedentary mice)[17,22]. These findings suggest that ELE may "prime" neuronal function to facilitate hippocampal learning. In this next experiment, we investigate whether ELE-enabled hippocampal memory is associated with altered presence of histone PTMs H4K8ac and H3K27me3 in regulatory regions of genes associated with learning and hippocampal plasticity; the prediction being that alterations of histone PTM occupancy as a result of ELE could promote gene expression programs necessary for facilitating memory consolidation. Wild type animals underwent ELE followed by OLM training on P42 and were sacrificed 60 min later (Fig. 6a). Dorsal hippocampi were dissected, and tissue was homogenized and processed for bulk RNA-seq. These data were compared to the ELE-induced DEGs, H4K8ac, and H3K27me3 occupancy occurring without learning (generated from Emx1-NuTRAP mice TRAP-seq and CUT&RUN-seq). We categorized DEGs based on the experimental group of wild type animals: ELE alone, ELE + 3 min learning stimulus, Sedentary+3 min learning stimulus, and Sedentary+10 min learning stimulus. To discover ELE-induced genes "primed" for regulation during memory consolidation, we focused specifically on cGAME (candidate genes associated with memory after exercise), cGAMES (candidate genes associated with memory after exercise or sedentary), and cGAMS (candidate genes associated with memory after sedentary) conditions (Fig. 6b, c and Supplementary Data 9). We then evaluated cGAME and cGAMES groups of genes for the presence or absence of histone modifications H4K8ac and H3K27me3 after ELE, suggesting addition or removal of these histone marks may promote a permissive (increased gene expression) or repressive (reduced gene expression) chromatin state, respectively (Fig. 6b′–c′ and Supplementary Data 10).

In the cGAMES group, there were 145 genes found to be significantly increased in the mice that underwent a threshold learning event (3 min for ELE; 10 min for sedentary mice). Of these 145 genes, 40% of them (58) had new H4K8ac following ELE (Fig. 6b–b′). We consider this group of genes to potentially be "primed," or readied, by ELE for rapid transcription to support hippocampal memory consolidation at 3 min OLM training, while also being involved in memory consolidation mechanisms in the threshold learning of sedentary mice (10 min OLM training). The cGAME group of genes are those which significantly increased in the ELE mice that underwent a threshold learning event but were not increased in any of the other groups (including the sedentary, threshold learning event/cGAMS group). These genes may be of unique importance for memory consolidation specifically occurring after ELE given their absence in sedentary learners. We found that the cGAME group had significant upregulation of 256 genes, with ~30% (76) of those genes receiving new H4K8ac following ELE alone (Fig. 6b–b′). We also evaluated the cGAMES and cGAME that were significantly downregulated during memory consolidation. We found 304 cGAMES and 361 cGAME genes (Fig. 6c). In both of these groups, a small minority of down-regulated genes had altered H3K27me3 after ELE, and this was mostly gain of H3K27me3 (cGAMES: 6.3%; cGAME: 6.4%; Fig. 6c′).

We next wanted to determine how ELE "priming" directed gene expression during memory consolidation that occurs after exercise (ELE 3 min OLM) relative to sedentary memory consolidation (Sed 10 min OLM). We compared these to the group that experienced the sub-theshold learning event (Sed 3 min OLM). We compared the relative log$_2$ fold changes (LFCs) of the genes in cGAME and cGAMES using z-scores for each group across the genes (Fig. 6d). In the vast majority of these primed genes, the ELE drove gene expression during 3 min of OLM in the same direction as 10 min of OLM training in sedentary animals (Fig. 6d). In some cases, the gene expression changes were more exaggerated in the same direction as a

sedentary threshold event (Fig. 6d). We interpret this to indicate that ELE enables a 3 min consolidation event to be threshold (when 10 min is required for a sedentary mouse) by priming these genes for altered gene expression in the same direction as a sedentary threshold event.

We next identified upstream regulators of cGAME and cGAMES genes with ELE-induced histone modifications, as these mediators are likely involved in long-term memory formation of a subthreshold learning event following ELE. The cGAMES and cGAME lists were evaluated using Qiagen's Ingenuity Pathway Analysis (IPA). First, candidate regulators were identified for the cGAME and cGAMES groups that have histone PTM changes as a result of ELE that we measured (Fig. 6e and Supplementary Data 11). Inhibition of MECP2 may indicate a reduction in DNA methylation. CREB1 is required for LTM[48,76]. SNCA may play a role in neuroplasticity by modulating synaptic vesicle transport[77].

Next, we wanted to identify gene networks that might be at play that are not just regulated by the histone PTMs we selected. Many more histone modifications exist and likely play a critical role in regulating gene expression during consolidation in addition to those we selected. We investigated the genes in cGAME and cGAMES using IPA and identified a set of likely upstream regulators for these genes taken together (Fig. 6e″ and Supplementary Data 11). Once again MECP2 inhibition and CREB1 activation appear to be indicated validating their presence in the previous IPA. Notably new to this group are inhibition of FMR1, KMT2D, and TCF20, along with activation of FASN, CTNNB1, and MKNK1. Next, we interrogated what networks might be unique to exercise-enabled consolidation by running the cGAME list through IPA analysis (Fig. 6e′ and Supplementary Data 11). Unique to this group, RTN4, an inhibitor of neurite outgrowth[78], was predicted to trend towards inhibition. GRIN3A, a glutamatergic NMDA receptor important to synaptic development and refinement[79], was predicted activated in this group. These findings suggest that ELE may enable consolidation by increasing neurite and synapse growth and development. TCF7L2 is also predicted to be activated. Given that this transcription factor is involved in Wnt signaling, and neurogenesis[80], and activated CTNNB1 is also predicted to be associated with threshold consolidation, altered Wnt signaling may be critical for ELE-enabled hippocampal memory consolidation.

## Discussion
Identifying gene regulatory mechanisms activated by exercise during sensitive developmental periods can aid in understanding how early-life exercise may promote a "molecular memory" of exercise[23,61], via the epigenome, that informs long-term cell function and behavioral outcomes. Here we present results demonstrating neuron-specific, simultaneous characterization of translating mRNA and associated histone modifications to reveal molecular signatures of the early-life exercise experience. To our knowledge, this is the first report describing use of the NuTRAP construct in a predominantly neuronal population. Additionally, our unique experimental approach for simultaneous INTACT and TRAP ("SIT") is a technical advance for NuTRAP applications: by modifying the nuclear chromatin isolation (INTACT) procedure to include cycloheximide, one can obtain both polyribosome-bound mRNA (using TRAP) and nuclear DNA from a single lysate to directly correlate gene expression programs with epigenetic regulatory mechanisms in the same set of cells. The field of neuroepigenetics has been challenged by limited approaches for coupling cell-type specific transcriptional profiles with chromatin modifications without compromising cell integrity (through flow sorting) or genomic coverage (through single-cell sequencing). Therefore, the NuTRAP model in combination

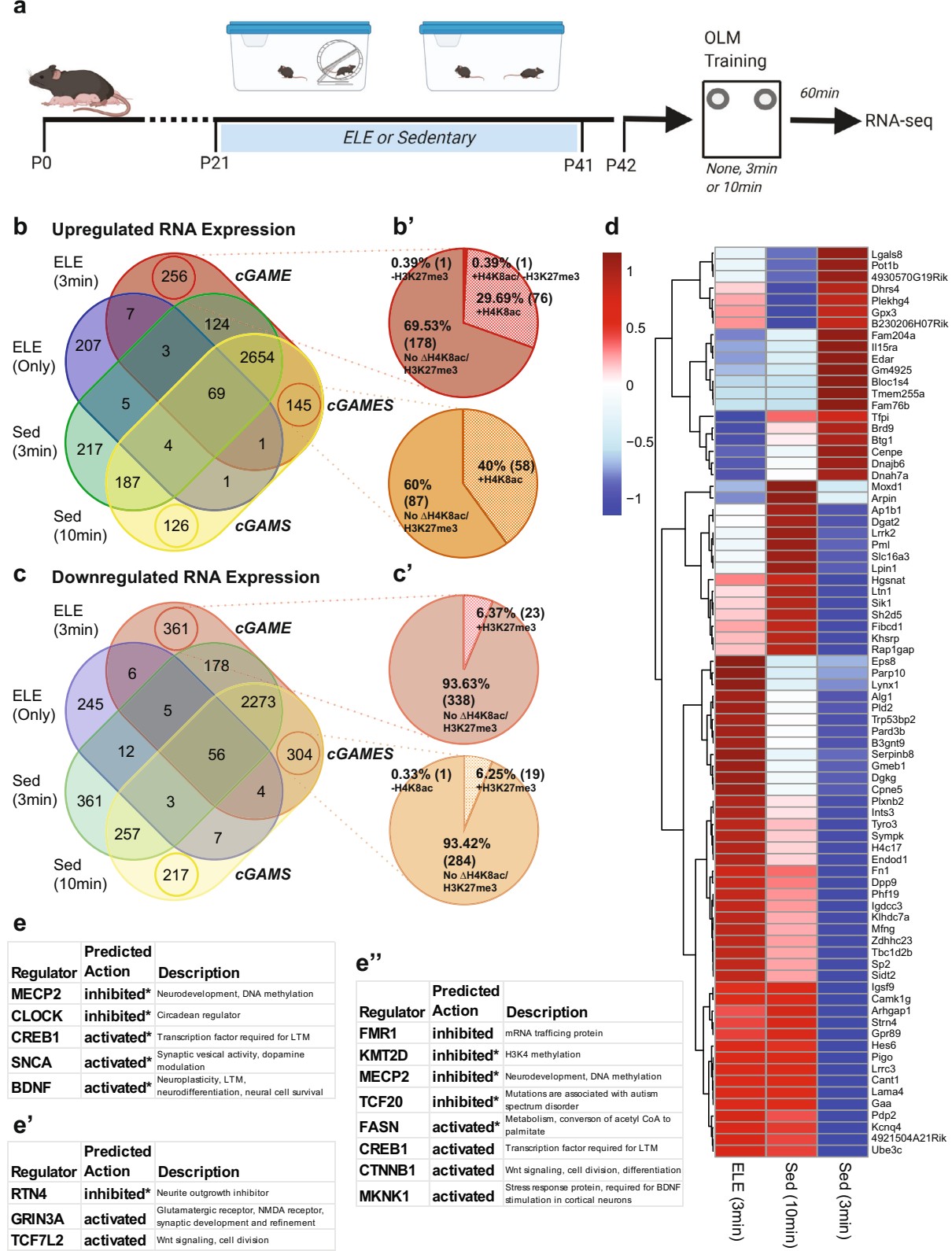

with the "SIT" protocol offers an exciting opportunity to overcome these challenges and perform paired transcriptomic and epigenomic analyses in any brain cell type of interest for which a Cre line exists[29,42].

This study focuses on the impact of exercise on hippocampal gene expression and associated epigenetic modifications specifically during juvenile-adolescent periods (the fourth through sixth postnatal weeks of life in mice). We discover neuron-specific gene expression alterations and associated changes in histone modifications H4K8ac and H3K27me3 occupancy after ELE. There have been recently published reports investigating the effects of voluntary exercise on adolescent and adult hippocampal gene expression and epigenetic modifications through sequencing methods from heterogenous cell populations[19,20], however this is

**Fig. 6 ELE may prime hippocampal gene expression supporting memory consolidation via alterations in H4k8ac and H3k27me3 occupancy.**
**a** Experimental design diagram. Created with BioRender.com. **b** Venn diagram representing upregulated genes relative to mice without ELE or learning in either ELE, ELE and 3 min OLM training, Sed and 3 min OLM training, and Sed and 10 min OLM training. **b'** Pie chart showing the relative percentage of genes that have new H4K8ac presence or H3K27me3 loss as a result of ELE in specific groups identified in the **b** labeled cGAME and cGAMES. **c** Same as **b** but for downregulated genes relative to mice without ELE. **c'** Pie chart of downregulated genes in the cGAME or cGAMES groups that have new H3K27me3 or new loss of H4K8ac as a result of ELE. **d** Z-score based heatmap of genes in the cGAMES and cGAME groups that have new H4K8ac or H3K27me3 presence or loss as a result of ELE. Expression values used to calculate the Z-scores were relative to mice that did not experience OLM training or ELE. **e** Upstream regulators identified by IPA of the genes represented by **d**. **e'** Upstream regulators identified by IPA of the genes in the cGAME (up and downregulated) group that are unique to that when only those are included in the IPA analysis. **e"** Upstream regulators identified by IPA of the genes in the cGAME and cGAMES (up and downregulated) group when the genes in those groups are put through IPA together. (asterisk indicates that the activated or inhibited indicator is registered from the IPAs based on the z-score for that regulator's expression and not directly indicated based on software prediction).

the first study to our knowledge that focuses on the effects of early exercise on the translatome and epigenome specifically within neurons. Our data revealed several notable genes with either chromatin-modifying or neurogenesis-regulating properties were either upregulated with new H4K8ac (*Prox1*, *Sntb2*, and *Kptn*), or downregulated with new H3K27me3 (*Col3a1*, *Efnb2*, *Epop*, and *Myoc*). One could consider the mechanistic implications for these genes: for example, ELE could "prime" the expression of *Prox1* by the addition of H4K8ac, and in turn, subsequent *Prox1* expression can promote neurogenic fate and cell location assignment[81]. ELE was also found to alter the expression of *Kptn*, which encodes a protein involved in actin dynamics and neuromorphogenesis[67]. Increased *Kptn* expression could promote cytoskeletal changes occurring with altered synaptic plasticity after exercise. Overall, the differentially expressed genes and their associated chromatin states occurring after ELE provide molecular candidates involved in the effects of ELE on hippocampal function.

Traditional epigenetic and gene expression sequencing approaches to bulk tissue samples are confounded by cellular heterogeneity. Further, current methods for paired analysis of the epigenome and transcriptome are restricted by the need for multiple transgenic animal models or the use of very expensive single-cell sequencing technologies. It is critical to obtain epigenomic information in a cell-type specific manner because there are unique epigenetic modifications distinguishing cell subtypes of the central nervous system[10,82]. The NuTRAP mouse was first developed by Roh et. al. to characterize adipocyte-specific and hepatocyte-specific epigenetic and transcriptional profiles[30]. The NuTRAP mouse has since been used to perform paired translatome and chromatin modification analysis in astrocytes and microglia, demonstrating its utility in CNS cell types[42]. The Emx1-NuTRAP mouse described here provides a rapid, cost-effective means for characterizing neuron-specific changes to epigenetic state and gene expression using whole-tissue homogenates. In our technical validation experiments, the "SIT" technique produced high quality RNA and DNA of sufficient concentration for downstream sequencing studies from a single mouse hippocampus[28]. This method effectively increases the total sample size used for each analysis as the tissue sample does not need to be separated for DNA and RNA isolation. CUT&RUN allows for low-input cell-specific chromatin analysis[38,62] further reducing the required amount of starting material. Indeed, the DNA and chromatin obtained from SIT could be used for other applications for evaluating chromatin landscape accessibility, such as ATAC-seq.

Presently, the field of neuroepigenetics is shifting to single-cell resolution for sequencing studies. Our study was performed in bulk tissue enriched for neurons expressing Emx1, which is limited by not distinguishing neuronal subtypes. As the goal of this work was to characterize the transcriptional and epigenomic effects of early-life exercise on neurons in general, cell subtype identity (as is often a priority in single-cell approaches) was not necessary for our initial hypotheses, but rather would offer a second layer of insight and future direction for this study. Single-cell approaches can bias against lowly-expressed genes, which may present the risk of not capturing small expression changes in rate-limiting genes that may have large changes in function resulting from ELE.

Given our previous findings that ELE can facilitate long-term memory and increase long-term potentiation[17], we hypothesized that this may be due to altering the epigenomic landscape to ultimately promote a transcriptional state that enables memory consolidation. This hypothesis is supported by the finding that in adult rodents, long-term memory is formed in response to a subthreshold learning event after exercise or HDAC3 inhibition and involves increased acetylation of neuroplasticity gene promoters[22]. This study identifies candidate gene expression networks and their epigenetic regulation that may be involved in the hippocampal memory benefits of ELE. We also identified a group of genes that are altered in their gene expression only in sedentary animals undergoing a threshold learning event ("cGAMS" group). The separation of this group from the cGAME and cGAMES groups indicates that ELE may engage a separate, perhaps complementary network of genes to promote memory consolidation. A limitation of this experiment was the use of RNA-seq data from wild type mice that underwent a hippocampal learning event to identify ELE-induced genes involved in memory consolidation. However, the genes we identified in the WT RNA-seq were also identified in the Emx1-NuTRAP exercised mice TRAP-seq data as "primed," i.e. having altered histone modifications after ELE without learning, thus implicating their role in memory consolidation. A next step will be to test the necessity and sufficiency of identified genes and gene networks, their histone modifications, and/or their upstream regulators mediating the effects of ELE on hippocampal memory performance. Another unanswered question is how long the memory-promoting effects of ELE last after exercise cessation, and whether there is stability of ELE-induced epigenetic modifications to allow for the persistence of ELE effects.

In summary, our work reveals ELE-induced transcriptomic and epigenomic signatures in hippocampal neurons using the Emx1-NuTRAP mouse for direct correlations between these data. A limitation for the use of the NuTRAP transgenic mouse is the requirement of a Cre line to exist for the experimenter's cell type of interest. In our Emx1-NuTRAP model, the Cre-recombinase in Emx1 expressing cells is constitutively expressed, and this may present off-target effects when present during development. Despite this, our study uses a feasible, previously undescribed approach in the field of neuroepigenetics to pair transcriptomic and epigenomic sequencing results from a population of hippocampal neurons. Another limitation of our study is the small sample sizes; however our Emx1-NuTRAP model is being continually applied in the lab and can be used for further comparisons and increasing correlations between data sets. Overall, our

approach revealed molecular epigenetic candidates for further mechanistic studies to examine the effects of early-life exercise on hippocampal function. Understanding how exercise uniquely impacts neuronal function during juvenile-adolescent developmental periods could provide insights into an early-life epigenetic memory of the ELE experience, revealing possible targets for improving memory function or mitigating functional deficits in the setting of early-life cognitive and neurodevelopmental disorders[83].

## Methods

### Resource availability

*Lead contact*. Further information and requests for resources and reagents should be directed to and will be fulfilled by the lead contact, Dr. Autumn Ivy (aivy@uci.edu).

*Materials availability*.

- No new reagents or materials were generated for this study.
- The mouse lines used for this study are available from Jackson Labs (Emx1-IRES-*Cre* knock-in mice, Jackson Laboratory Stock No: 005628, and NuTRAP mice, Jackson Laboratory Stock No: 029899).

*Animals*. Emx1-IRES-*Cre* knock-in mice (Jackson Laboratory Stock No: 005628), NuTRAP mice (Jackson Laboratory Stock No: 029899), and C57Bl6/J (Jackson Laboratory Stock No: 000664) wild type mice were obtained from Jackson Laboratories. Female Emx1-IRES-*Cre* and male NuTRAP mice were crossed to generate the Emx1-NuTRAP mice used for this study. Mice were given free access to food and water, and lights were maintained on a standard 12-h light/dark cycle. On postnatal day (P) 21, mice were weaned and pair-housed in either standard bedding cages or cages equipped with a running wheel. Only male mice were used for the studies performed. Experiments were conducted according to US National Institutes of Health guidelines for animal care and use and were approved by the Institutional Animal Care and Use Committee of the University of California, Irvine.

*Behavior*. Wild type mice were habituated, handled and then underwent OLM training[84] for either 3 or 10 min at P41 as described previously[17]. Briefly, on postnatal day 36, mice were brought into a testing room with reduced room brightness (LUX 8.5-11). Mice were handled for ~2 min each, for a total of 5 consecutive days prior to the OLM training session (twice a day for the first 2 days followed by once a day for the next 3 days). Habituation sessions were 5 min, twice per day for 3 days (P39-P41) and occurred within chambers containing four unique spatial cues on each wall of the chamber (horizontal lines, black X, vertical strip, and blank wall). Habituation overlapped with the last 3 days of handling. During the training phase (P42), mice were placed into the same chambers with two identical objects and exposed to either a subthreshold (3 min) or threshold (10 min) training period. Mice were sacrificed for consolidation experiments 1 h later.

*Exercise paradigm*. After weaning on P21, male mice from the same litters were randomly divided and pair-housed in either standard cages without a wheel or cages equipped with a single, stainless-steel running wheel[17]. Animals housed with running wheels had free access to the wheels from P21–41. Real-time data acquisition (Vital View Software) was used to track distance ran recorded by magnetic sensors that detect wheel revolutions. Each monitored wheel tracked running distance for the entire cage (two mice per cage). Revolutions were quantified for every minute daily for the duration of the exercise period and converted to distance ran per cage (km). Exercised mice that underwent OLM training were individually tracked as described in Valientes et al. 2021[85].

*Simultaneous INTACT and TRAP ("SIT")*. On P42 both ELE and sedentary control Emx1-NuTRAP mice were sacrificed by rapid cervical dislocation and both hemispheres of the hippocampus were dissected out and collected for tissue processing. Hippocampal tissue samples were mechanically homogenized in 1 mL of nuclear preparation buffer (NPB) (10 mM HEPES (pH 7.5); 1.5 mM MgCl2; 10 mM KCl; 250 mM sucrose; 0.1% IGEPAL CA-630; 0.2 mM DTT; 100 μg/mL cycloheximide; 1x Complete EDTA-free protease inhibitor (Roche)) by razor blade. Samples were then triturated with an 18 G needle and syringe 50x on ice.

Homogenized tissue samples were pipetted slowly onto 100 μm cell strainer caps on 5 mL tubes and centrifuged at 1000rcf for 10 min at 4 °C. Cell pellets were resuspended, and samples were pipetted onto 40 μm cell strainer caps on 5 mL tubes and centrifuged at 1000rcf for 10 min at 4°C. For nuclear isolation, 100 μL of Streptavidin-coated Dynabeads were washed twice with 1 mL NPB and resuspended with 100 μL NPB. Cell pellets were resuspended and added to washed Streptavidin-coated Dynabeads. Samples were briefly pipetted to thoroughly suspend the beads and incubated on ice for 20 min. Samples were placed on a magnetic stand to pull down nuclei bound to the beads. The supernatant from each

sample was slowly pipetted off the beads and transferred to new 1.5 mL tubes for RNA isolation (described later). Nuclei-bound beads were washed twice with 1 mL INTACT buffer (10 mM HEPES (pH 7.5); 1.5 mM MgCl2; 10 mM KCl; 250 mM sucrose; 0.1% IGEPAL CA-630; 1x Complete EDTA-free protease inhibitor (Roche)) and resuspended in 100 μL INTACT buffer. Bead-bound nuclei were flash frozen on dry ice and stored at −80 °C until ready for CUT&RUN chromatin digestion and DNA purification.

For RNA isolation, 100 μL of concentrated TRAP isolation buffer (10 mM HEPES (pH 7.5); 117 mM MgCl2; 1 M KCl; 10% IGEPAL CA-630; 100 μg/mL cycloheximide; 11 mg/mL sodium heparin; 20 mM DTT; 2.2units/μL Rnasin; 1x Complete EDTA-free protease inhibitor (Roche)) was added to each sample (supernatant above). Samples were mixed by gentle pipetting and centrifuged at 16,000rcf for 10 min at 4 °C. The supernatant from each sample was transferred to new 1.5 mL tubes and 50 μL of protein-G Dynabeads was added to each sample. (To bind anti-GFP, 100 μL of protein-G Dynabeads were washed twice with 500 μL NPB and resuspended to a final 100 μL volume. 2 μL anti-GFP antibody (Sigma-Aldrich, G6539, Lot:128M4867V) was added to each tube and incubated for 1 hour on a rotating nutator at 4°C. Excess anti-GFP was washed off with 200 μL NPB and beads were resuspended in a final 100 μL volume.) Samples were incubated for 3 hours on a rotating nutator at 4 °C. Beads were washed 3x in 1 mL high salt wash buffer (HSWB) (50 mM Tris (pH 7.5); 12 mM MgCl2; 300 mM KCl; 1% IGEPAL CA-630; 2 mM DTT; 100 μg/mL cycloheximide) at 4 °C. After the third wash the beads were slowly resuspended in 350 μL RLT buffer (from Qiagen RNeasy Mini kit #74004) by slowly pipetting up and down and incubated on a rotating nutator for 30 min at room temperature. Samples were placed on a magnetic stand to pull down the beads, and the supernatant from each were transferred to new 1.5 mL tubes and stored at -80°C until ready for RNA purification.

*Separate TRAP RNA Isolation*. On P42, mice (*n* = 3 mice/group) were sacrificed and the hippocampus was dissected out as described above. Either the left or right brain hemisphere of the hippocampus was processed for TRAP RNA isolation[29,86] and the other hemisphere was processed for INTACT nuclear isolation (see below). Hippocampal tissue was homogenized with a motorized pestle in 1 ml of TRAP homogenization buffer (50 mM Tris, (pH 7.5); 12 mM MgCl2; 100 mM KCl; 1% NP-40; 100 μg/ml cycloheximide; 1 mg/ml sodium heparin; 2 mM DTT; 0.2units/μl RNasin; 1x Complete EDTA-free protease inhibitor (Roche)). Samples were centrifuged at 16,000rcf for 10 min at 4 °C. The supernatant from each sample was collected and incubated with 50 μL of protein G Dynabeads (washed twice in 1 mL TRAP homogenization buffer without RNasin or protease inhibitor and resuspended in 50 μL of the same buffer). Samples were incubated for 30 min on a rotating nutator at 4 °C. Following the incubation, 100 μL of anti-GFP bound protein-G Dynabeads was added to each sample. (To bind anti-GFP, 100 μL of protein-G Dynabeads were washed twice with 500 μL TRAP homogenization buffer without RNasin or protease inhibitor and resuspended to a final 100 μL volume. 2 μL anti-GFP antibody (Sigma-Aldrich, G6539, Lot:128M4867V) was added to each tube and incubated for 1 hour on a rotating nutator at 4°C. Excess anti-GFP was washed off with 200 μL TRAP homogenization buffer without RNasin or protease inhibitor and beads were resuspended in a final 100 μL volume.) Samples were incubated for 3 h on a rotating nutator at 4 °C. Beads were washed 3x in 1 mL HSWB at 4 °C. After the final wash, the beads were resuspended in 350 μL RLT buffer (from Qiagen RNeasy Mini kit #74004) and incubated on a rotating nutator for 30 min at room temperature. Following the incubation, samples were transferred to new 1.5 mL tubes and stored at −80 °C until ready for RNA purification.

*Separate INTACT nuclear isolation*. The remaining hippocampal hemisphere from each mouse (not used for TRAP RNA isolation above) was processed for INTACT DNA isolation using methods similar to those described[29,87]. Tissue samples were quickly chopped with a razor blade, added to 1 mL of INTACT buffer, and triturated with an 18 G needle and syringe 50x on ice. Samples were then added onto 100 μm cell strainer caps on 5 mL tubes and centrifuged at 1000rcf for 10 min at 4 °C. Cell pellets were resuspended and samples were pipetted onto 40 μm cell strainer caps and centrifuged at 1000rcf for 10 min at 4 °C. In all, 100 μL of streptavidin coated Dynabeads (washed twice with 1 mL INTACT buffer and resuspended to a final 100 μL volume) were added to each sample, and samples were incubated on ice for 20 min. After incubation, nuclei bound beads were washed twice with 1 mL and resuspended in 100 μL INTACT buffer. Bead-bound nuclei were flash frozen on dry ice and stored at −80 °C until ready for CUT&RUN chromatin digestion and DNA purification.

*RNA purification and sequencing library preparation*. RNA was purified using the Qiagen RNeasy kit according to the manufacturer's protocol. After purification, RNA quantity and quality were checked using the Qubit RNA High Sensitivity assay and Agilent Bioanalyzer's Eukaryote Total RNA Pico assay, respectively, at the University of California, Irvine Genomics High Throughput Facility (UCI GHTF). Using an input of 15 ng of total RNA, mRNA was isolated using NEXTFLEX Poly(A) Beads 2.0, and the mRNA sequencing libraries were generated using the NEXTFLEX Rapid Directional RNA-Seq Kit 2.0 according to the manufacturer's instructions. Final libraries were sent to the UCI GHTF for Qubit dsDNA High Sensitivity assay and the Agilent Bioanalyzer DNA High Sensitivity assay to determine the quantity and quality of the RNA-seq libraries, respectively. Libraries were sequenced at the UCI

GHTF using 100 bp paired end reads on a single lane of an Illumina NovaSeq6000 to a minimum sequencing depth of 50 million reads.

*Hippocampal mRNA Isolation.* P42 wild type male mice were sacrificed by cervical dislocation. Dorsal hippocampus was extracted by bisecting the isolated hippocampus. The tissue was homogenized and RNA was extracted using the Qiagen RNeasy kit according to the manufacturer's protocol. Sequencing libraries were prepared from the RNA as described under the section "RNA purification" and the UCI GHTF performed sequencing library preparation, except that ERCC ExFold RNA Spike-In Mixes (ThermoFisher cat. 4465739) were added according to the manufacturer's instructions.

*Cleavage under target and release using nuclease (CUT&RUN).* For chromatin analysis, CUT&RUN was performed using the EpiCypher CUTANA CUT&RUN protocol with slight modifications similar to what has already been described[38]. Nuclei bound to beads were thawed and pulled down on a magnetic stand. The supernatant was removed and beads were resuspended in 200 μL wash buffer (20 mM HEPES (pH 7.5); 150 mM NaCl; 0.5 mM spermidine; 1x Complete EDTA-free protease inhibitor (Roche)). Each sample was divided into three aliquots (50 μL for anti-H3K27me3 (Cell Signaling Technologies, C36B11, Lot: 16), 100 μL for anti-H4K8ac (Epicypher, 13-0047, Lot: 20202001-11), and 50 μL for anti-IgG control (Rabbit IgG Fisher Scientific, 026102)). Beads were pulled down on a magnetic stand, resuspended in 100 μL wash buffer and incubated for 10 min at room temperature. After incubation the beads were pulled down, the supernatant was removed, and the beads were resuspended in 50 μL ice cold antibody buffer (20 mM HEPES (pH 7.5); 150 mM NaCl; 0.5 mM spermidine; 0.001 mM digitonin; 2 mM EDTA; 1x Complete EDTA-free protease inhibitor (Roche)). In all, 0.5 μL of the appropriate antibody was added to each sample, and the samples were mixed by gentle pipetting prior to overnight incubation at 4 °C on a nutator.

On the second day, the beads were pulled down on a magnetic stand, the supernatant was removed and the beads were washed twice in 250 μL ice cold wash buffer with 0.001% digitonin. After the washes, beads were resuspended in 50 μL ice cold wash buffer with 0.001% digitonin, and 2.5 μL of CUTANA pAG-MNase was added to each sample and mixed by gentle pipetting. Samples were incubated for 10 min at room temperature and beads were pulled down on a magnetic stand. The supernatant was removed, and the beads were washed twice in 250 μL ice cold wash buffer with 0.001% digitonin and resuspended in 50 μL of the same. In total, 1 μL of 100 mM CaCl₂ was added to each sample and mixed by gentle pipetting. Samples were incubated for 2 h on a nutator at 4 °C. Following the incubation, 33 μL of stop buffer (340 mM NaCl; 20 mM EDTA; 4 mM EGTA; 50 μg/mL RNase A; 50 μg/mL Glycogen) and 1.65 μL of Spike-In DNA (Cell Signaling Technology, #40366) was added to each sample. Samples were incubated in a preheated thermocycler for 10 min at 37 °C. After incubation, samples were transferred to new 1.5 mL tubes and immediately processed for DNA purification.

*DNA purification and library preparation.* DNA was purified using the Monarch PCR and DNA cleanup kit (New England Biolabs, 13-0041) according to the manufacturer's instructions. After purification, DNA fragments were used to generate sequencing libraries using the NEXTFLEX Rapid DNA-Seq Kit 2.0 according to the manufacturer's instructions. Indices were diluted to a final concentration of 1:1000 before being added to samples, and a total of 13 PCR cycles was used for DNA amplification of the libraries. Final libraries were sent to the UCI GHTF for Qubit dsDNA High Sensitivity assay and the Agilent Bioanalyzer DNA High Sensitivity assay to determine the quantity and quality of the CUT&RUN-seq libraries, respectively. Libraries were sequenced at the UCI GHTF using 100 bp paired end reads on a single lane of an Illumina NovaSeq6000 to a minimum sequencing depth of 10 million reads.

*qPCR.* cDNA was generated from TRAP-isolated and total input RNA samples ($n = 2$ samples/group) using the Transcriptor First Strand cDNA Synthesis Kit (Roche) following the manufacturer's instructions. Relative gene expression for *Aqp4* (F: 5′-ATCCAGCTCGATCTTTTGGA -3′, R: 5′-TGAGCTCCACATCAGGACAG -3′), *Tubb3* (F: 5′-GTCTCTAGCCGCGTGAAGTC -3′, R: 5′-GCAGGTCTGAGT CCCCTACA -3′), *Cd11b* (F: 5′-CCCATGACCTTCCAAGAGAA -3′, R: 5′-ACACT GGTAGAGGGCACCTG -3′), and *Mog* (F: 5′-AAGAGGCAGCAATGGAGTTG -3′, R: 5′-GACCTGCAGGAGGATCGTAG -3′) was determined by qPCR using FastStart Essential DNA Green Master Mix (Roche, 06402712001) following the manufacturer's protocol. Cycle counts for mRNA quantification were normalized to *Gapdh* (F: 5′-CGTCCCGTAGACAAAATGGT -3′, R: 5′-GAATTTGCCGTGAGTGGAGT -3′). Quantification was performed using the Pfaffl method[88].

*Immunofluoresence.* Emx1-NuTRAP mice were anesthetized intraperitoneally with 50 mg/kg of sodium pentobarbital and transfused transcardially with 4% paraformaldehyde (PFA) in 1x PBS. After perfusion, whole brains were dissected out and incubated in 4% PFA overnight at 4 °C. After fixation, brains were incubated in 30% sucrose for 72 hours and mounted in optimal cutting temperature (OCT) compound. 50μm coronal sections were sliced through the hippocampus using a microtome, and every tenth section was collected in 1 mL of cryoprotectant and stored at -20 °C until ready for immunofluorescent labeling. For immunofluorescence, sections were brought up to room temperature and washed 2x in 1x

PBS on a nutator for 5 min each. Sections were then washed 3x in 1x PBS with 0.3% triton-X. After washes, sections were incubated with mCherry polyclonal antibody (ThermoFisher, PA5-34974, Lot: UG2804409F) at a 1:200 dilution in blocking solution (1x PBS with 1% BSA) overnight at 4 °C on a nutator. The following day, sections were washed 3x in 1x PBS. Sections were incubated for two hours at room temperature with goat anti-rabbit Alexa Fluor 594 secondary antibody (Invitrogen, A-11012, Lot: 2119134) at a 1:200 dilution in blocking solution. After incubation, the sections were washed twice in 1x PBS. To label nuclei, sections were incubated with DAPI at a 1:10,000 dilution in 1x PBS for 20 min. Sections were then washed 3x in 1x PBS and cover slipped with Vectatshield®. Fluorescent images were acquired using a Keyence BZ-X810 All-in-One Fluorescence microscope using the optical sectioning module.

*Fluorescence activated cell sorting.* Fluorescence activated cell sorting (FACS) was performed to characterize neuronal and astrocytic NuTRAP cassette expression. Whole hippocampal tissue from Emx1-NuTRAP mice was isolated and single-cell suspensions were immunostained for cytometric analysis[89]. We used antibodies for THY-1 (OX7) AlexaFluor™ 647 (Santa Cruz Biotechnology, sc-53116 AF647, Lot: F2716; concentration: 1:50) and S100β (Abcam, ab41548, Lot:GR3326165-1; concentration: 1:200) with an AlexaFluor™ 405 goat anti-rabbit IgG (H + L) (Invitrogen, A31556, Lot: 2273716; concentration:1:800) secondary antibody. FACS analysis was performed using a BD FACSAria™ Fusion Flow Cytometer (BD Biosciences) at the University of California, Irvine Stem Cell Core. Samples and single-stain controls were analyzed using the FlowJo v10.8.1 software (BD Biosciences). Samples and controls were positively gated for live cells (SSC-A and FSC-A) and single cells (FSC-H and FSC-A), and negatively gated for autofluorescence (Comp-Alexa Fluor 647-A (Thy1) and Comp-BV421-A (S100β)). Fraction of GFP + neuronal cells was determined using quartile analysis of Thy-1 and Comp-GFP-A (GFP). Fraction of GFP + astrocytic cells was determined using quartile analysis of S100β and Comp-GFP-A (GFP).

*Statistics and reproducibility*
Bioinformatics analysis: *Reference assembly*: For all sequencing analysis that required it, the primary reference assembly used was Genome Reference Consortium Mouse Build 38 patch release 6 (GRCm38.p6).

*RNA-seq analysis:* FastQ files were quality checked for sequencing errors using FastQC (version 0.11.9)[90]. No files were found to have sufficient quality errors to discount their use. Files were aligned using STAR Aligner (version 2.7.3a)[91]. Duplicate reads were removed using Picard Tools (version 1.87)[92]. SAM Tools was used to convert BAM files to SAM files for use in downstream analysis. FastQC, alignment and duplicate removal were all preformed on the High-Powered Compute Cluster (HPC3) operated by The Research Cyberinfrastructure Center (RCIC) at the University of California, Irvine. R (version 4.1.0)[93] was used for differentially expressed genes (DEG) analysis. Genomic Alignments (version 1.28.0)[94] (summarizeOverlaps mode = "IntersectionNotEmpty", singleEnd=FALSE, ignore.strand=FALSE, fragments=TRUE), Genomic Features (version 1.44.0) (exons by gene), and R SAM Tools (version 2.8.0)[95] (yieldSize=100000) were used to extract a count matrix and generate a summarized experiment object. DEseq2 (version 1.32.0)[96] was used to perform a DEG analysis. DEseq2's default negative binomial distribution was used to model RNA-seq counts and the default Wald test method was used to test the hypothesis that a gene was differentially expressed. DEseq2's interpretation of the Benjamini–Hochberg test for multiple corrections was used to produce the padj values indicated in our DEG tables. A less stringent cutoff was used for identification of significant genes to identify broad patterns: absolute value of the Log2 fold change greater than 0.3875 and a Wald p value of less than 0.05. Ensemble IDs were converted to gene symbols using BiomaRt (version 2.48.1)[97,98]. PCA plots using the top 2000 genes were generated using ggplot2 (version 3.3.4)[99]. Samples were determined as outliers if the variability between the samples heavily weighted PC1 to those samples. This excluded 2 sedentary samples and 1 ELE sample. Heatmaps were generated using gplots (version 3.1.1)[100] and RColorBrewer (version1.1-2)[101]. Volcano Plots were generated using ggplot2[99]. Venn diagrams were generated using the Venn diagram tool from Bioinformatics and Evolutionary Genomics[102]. Gene ontology analysis was performed using Panther Classification System (version 16.0)[46]. Genes were also categorized and a leading edge heatmap was also generated using Gene Set Enrichment Analysis (version GSEA 4.1.0)[57]. Upstream regulators were identified from the DEGs upregulated by ELE using Qiagen's Ingenuity Pathway Analysis (IPA) (Fall 2021 Release)[47].

*Likelihood ratio test:* to determine which genes had a significant effect of hemisphere we performed DEseq2's Likelihood Ratio Test on the RNA-seq data from the exercised or sedentary, left and right hemisphere separately isolated samples. The full model used for DEseq2's full comparison was "hemisphere + exercise + hemisphere:exercise" and the reduced models were either "hemisphere", "exercise", "exercise + exercise:hemisphere", or "hemisphere + exercise:hemisphere". This data is presented in Supplementary Data 5.

*2-way ANOVA:* We also used a 2-way ANOVA to further support the analysis of the effect of hemisphere. This analysis used R's aov function set for the two way ANOVA function. This analysis was run individually on each gene and presented in a table (Supplementary Data 5).

*CUT&RUN-seq analysis:* FastQ files were quality checked for sequencing errors using FastQC (version 0.11.9)[90]. No files were found to have sufficient

quality errors to discount their use. FastQ files were aligned using Bowtie2 (version 2.4.1)[103]. Significant peaks were called from these aligned files using SEACR, developed in CUT&Tag for efficient epigenomic profiling of small samples and single cells[62,103], according to CUT&Tag Data Processing and Analysis Tutorial (updated August 12 2020): https://www.protocols.io/view/cut-amp-tag-data-processing-and-analysis-tutorial-bjk2kky. An FDR higher than 0.1 was considered too high for called peaks. This threshold excluded 4 samples (3 ELE H4K8ac separate isolation samples and 1 ELE H4K8ac SIT sample). Peaks were called using the stringent and 0.01 settings. Significant peaks were annotated using ChIPseeker (version 1.8.6)[104] using TxDb.Mmusculus.UCSC.mm10.knownGene[105] as a reference with the following settings: tssRegion = c(-3000, 3000), TxDb = TxDb.Mmusculus.UCSC.mm10.knownGene, level = "transcript", assignGenomicAnnotation = TRUE, genomicAnnotationPriority = c("Promoter", "5UTR", "3UTR", "Exon", "Intron", "Downstream", "Intergenic"), annoDb = NULL, addFlankGeneInfo = FALSE, flankDistance = 5000, sameStrand = FALSE, ignoreOverlap = FALSE, ignoreUpstream = FALSE, ignoreDownstream = FALSE, overlap = "TSS", verbose = TRUE). This approach annotated peaks to the closest gene by distance from the promoter, except if that peak falls within a gene before the distal intergenic region. Ensemble IDs were converted to gene symbols using BiomaRt[97,98]. Gene lists of peaks present in each condition were used to generate Venn diagrams, as with RNA-seq analysis[102]. Further Venn diagrams were generated comparing DEGs with peak calls. Peaks were visualized using UCSC Genome Browser and Genome Browser in a Box[106–108].

Normalized counts (for read depth) at binned sections of the DNA as well as RNA-seq gene expression count (normalized for read depth) were compared between simultaneous and separate isolations as well as left and right hemispheres using Spearman's correlations in R with the package ggpubr (version 0.4.0)[109]. Bar plots were generated using Graph Pad 9 PRISM.

## Data availability

RNA-, TRAP- and CUT&RUN-seq data is deposited in the Gene Expression Omnibus (GEO) and is publicly available as of the date of publication. Accession numbers: GSE208715 (all datasets), GSE208633 (CUT&RUN-seq data), and GSE208714 (RNA- and TRAP-seq data). Any additional information required to reanalyze the data reported in this paper is available from the lead contact upon request.

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

## Acknowledgements
This work was supported by the National Institute of Neurological Disorders and Stroke of the National Institutes of Health, under award number K12NS098482, the Conte Center @ UCI Seed Grant, UC Irvine Institute for Clinical and Translational Sciences Pilot Award, and the Robert Wood Johnson Foundation Amos Medical Faculty Development Program (awarded to A.S.I.). We thank the UC Irvine School of Medicine, Departments of Pediatrics for support. We would also like to thank Drs. Melanie Oakes and Jenny Wu of the UC Irvine Genomics Research and Technology Hub for next generation sequencing experiments and bioinformatics support.

## Author contributions
A.M.R., T.D.F., N.E.N., and A.S.I. designed the studies and interpreted the results. A.M.R., T.D.F., and A.S.I. performed INTACT and TRAP studies. A.M.R. built libraries for RNA- and TRAP-sequencing. N.E.N. performed CUT&RUN-seq experiments and built libraries for sequencing. N.E.N. and A.S.I. performed flow cytometry experiments. A.M.R. and A.S.I. performed immunostainings and immunofluorescent imaging. D.A.V. and A.B. built exercise cages and monitored running behavior. A.M.R. performed all bioinformatic analysis of sequencing data. A.M.R. and N.E.N. analyzed sequencing data, performed GO, GSEA, and IPA analyses, and all other downstream analyses except where noted. A.S.I. performed behavioral studies. T.D.F. performed Spearman's correlations and gene length by fold change analysis. T.D.F. performed qPCR. N.E.N. designed the figures. A.M.R., T.D.F., and A.S.I. wrote and edited the manuscript, with N.E.N. providing edits to the manuscript. All authors discussed the results and edited and approved the manuscript.

## Competing interests
The authors declare no competing financial interests.
