## [Peer Review File · Communications Biology]

Reviewers' comments:

Reviewer #1 (Remarks to the Author):

The authors examine effect of early life exercise on neuronal histone patterns and gene expression which is an interesting area of investigation. They also present a 'simultaneous' INTACT and TRAP which would have advantage of using smaller sample amounts and potentially give clearer correlations between epigenomic and translatomic patterns as data would be from the same cellular population. Overall this is an interesting area of investigation but the presentation is a bit diffuse with the principle biological implications requiring some clearer presentation for the reader. Some technical issues were also identified.

Figure 1 and Supplemental Table 1 – Some clarification on the flow cytometry analysis is needed. A gating of GFP+ cells for Thy+, S100b+, and double negative would aid in understanding the specificity of the Emx1 cre labeling. While there is a good population of GFP+/Thy1+ - 15% of singlets, there is also a substantial portion of GFP+/Thy1- singlets (8%). With the caveat that neuronal cell surface labeling for flow cytometry is notoriously difficult it would be very helpful to understand what that substantial Thy1- population is. CD24 could be another neuronal marker to try.

The RNA-Seq against markers is very supportive of specificity but it would be good to also look more broadly at cell enrichment lists generated by Zhang and Barres labs (e.g., PMID: 29892006) in addition to selected marker genes. This gives a broader insight and is helpful for the field. This could easily be a supplemental figure.

In the comparison of the simultaneous and separate TRAP the limited overlap of DEGs identified (Figure 2F) is a bit concerning. This could be a statistical power issue arising from the different sample numbers (2,3 for simultaneous and 3,3 for TRAP). Perhaps a plot comparing the fold change between the two methods would help address this question of whether the different in numbers are a reflection of genes that just pass or just fail to pass statistical criteria. It is also not clear what exact statistical tests and multiple testing corrections are used in the RNA-Seq analysis.

An advantage of the simultaneous isolation protocol should be that there is stronger correlation between epigenomic and translatomic profiles in the simultaneous vs separate protocols. Can the authors compare the cut & run vs expression relationships (Fig 2) in the two methods to address this? Figure 3A - As described above, more detail on the statistical approach is needed here. Is the Venn a comparison of independent t-tests performed on each hemisphere? With the small sample sizes (n=2-3/group) and thus the low power, presumably after multiple testing correction, it is easy to get genes that barely pass the critical threshold in one comparison but just fail to reach the p-value cutoff in the other. To really get at this central question of laterality it would seem that a 2-Way ANOVA design is needed and the interaction term of ELE and Hemisphere would provide an unambiguous determination of laterality. This will give the clearest indication of differences in DEGs by hemisphere.

The multiple way Venn diagrams (Fig 5A & A') could benefit from an upset plot presentation (<https://r-graph-gallery.com/upset-plot.html>). This would aid the reader in understanding the overlaps between sets.

A weakness in the overall study design in that for the ELE analysis in Figure 6 the analysis switches back to whole tissue analysis from TRAP and INTACT of a specific cell type. This reduces the interoperability of the data with the other experiments.

Reviewer #2 (Remarks to the Author):

Raus et al have studied how adolescent engagement in aerobic exercise translates to improved hippocampal-dependent memory, and propose that it involves improved transcriptional dynamics through histone modifications. To facilitate this study, they describe an innovative approach to isolating mRNA and chromatin from excitatory hippocampus neurons using Emx1-NuTRAP transgenic mice. This novel protocol is worth publishing (Nature Methods or Nature Protocols), but the biological

interpretations regarding transcriptional responses during memory consolidation are weak.

Major points of concern:

The authors have described the SIT protocol well. They have performed adequate controls. The absence of β in their initial cell sorting study significantly boosts the utility of this novel technique. One concern/query I have is regarding the concentration of TRAP-isolated mRNA yielded, which appears to be rather low <15ng/uL. Less specific methods of total RNA isolation typically yields >300ng/uL for bilateral whole hippocampus tissue. Where is all the excess RNA lost to? Is it expected that nuclear/ribosome-associated mRNA represents such a small fraction of the total RNA pool in neurons?

The individual yields for (TRAP/INTACT) SIT indicate that 5 out of 8 samples yielded 15+ng/uL. The consistency and efficiency of the process is arguably not optimal. Is this reflective of issues with methodology or samples?

Five out of twelve samples of TRAP only isolation yielded more than the reported average 7+ng/uL, there was one sample that yielded 23ng/uL. There is also a great degree of variance in this data set too, which likely reflects the sensitivity of the technique, but also raises concerns about the quality of this data set and the appropriateness of claiming that TRAP alone yielded ~half of the SIT protocol consistent with using one hemisphere vs bilateral samples. These efficiency issues are critical to new protocols since it would greatly affect the design and reproducibility of data for future studies relying on this method, if published.

No objective measure of memory consolidation is provided, so the authors cannot claim any benefit of exercise on memory capacity. If anything, the subsequent molecular responses only reflect response to a novel environment or experimenter handling.

The emphasis on "early life" provision of exercise is lost due to the absence of an aged exercise group. The claims to 'transcriptional programs by ELE that could be unique to exercise timing..' is rather pre-emptive.

Figure 4 title 'ELE leads to... during adolescence'. The mice were killed at 42 days of age. Since behavioural testing and molecular analysis were performed on 6 week old tissue, this is reflective of young adult, not the adolescent brain.

Minor points:

It was confusing for the initial SIT RINs reported in lines 149-150 to be stated as >8, but then an average was later reported in line 200.

It is a shame that only male mice were studied.

Were there one or two running wheels per cage? Was there competition for the running wheels, as this could contribute to stress, jostling for dominance to access the wheels?

It would be appropriate to report brightness of behaviour rooms in LUX, rather than simply reduced lighting. Line 734.

I feel the subsection headings do not draw sufficient attention to the main study of sedentary vs exercise conditions.

The finding of transcriptional symmetry between L vs R hippocampus is an important confirmation.

However the authors only compared L vs R for the ELE group., It would be appropriate to first demonstrate the presence of transcriptional symmetry in the sedentary controls. The reported transcriptional response to exercise is largely agreeable with the literature.

It is not clear how the up-regulation of expression can be concurrent with an active histone mark AND a repressive mark for 14 genes (Results Line 408). Is there evidence in the literature for this seemingly conflicting occurrence?

Reviewer #1 (Remarks to the Author):

The authors examine effect of early life exercise on neuronal histone patterns and gene expression which is an interesting area of investigation. They also present a 'simultaneous' INTACT and TRAP which would have advantage of using smaller sample amounts and potentially give clearer correlations between epigenomic and translational patterns as data would be from the same cellular population. Overall this is an interesting area of investigation but the presentation is a bit diffuse with the principle biological implications requiring some clearer presentation for the reader. Some technical issues were also identified.

We thank the reviewer very much for their interest in the study and appreciate the critiques provided. We believe that the feedback provided here greatly enhance the manuscript and specifically our presentation of the data. We have revised parts of the text (specific examples are in subsequent reviewer responses) to emphasize the principle biological implications and aid in clarity of our conclusions.

- 1. Figure 1 and Supplemental Table 1 – Some clarification on the flow cytometry analysis is needed. A gating of GFP+ cells for Thy+, S100b+, and double negative would aid in understanding the specificity of the Emx1 cre labeling. While there is a good population of GFP+/Thy1+ - 15% of singlets, there is also a substantial portion of GFP+/Thy1- singlets (8%). With the caveat that neuronal cell surface labeling for flow cytometry is notoriously difficult it would be very helpful to understand what that substantial Thy1- population is. CD24 could be another neuronal marker to try.*

We appreciate this request for clarification. We have now included our full flow cytometry gating scheme in the supplemental information (Supplementary Figure 2) for Thy1+, S100b+ and double negatives. Regarding the GFP+/Thy1- singlets, we interpret this finding to reflect Emx1 labeling of dividing immature granule cells, as Thy1 is not expressed in granule cells of the DG until they are 4-5 weeks old (see references below). We have now addressed this point in the text of the manuscript (Lines 130-133).

<https://www.ncbi.nlm.nih.gov/pmc/articles/PMC6672603/>

<https://pubmed.ncbi.nlm.nih.gov/22533643/>

- 2. The RNA-Seq against markers is very supportive of specificity but it would be good to also look more broadly at cell enrichment lists generated by Zhang and Barres labs (e.g., PMID: 29892006) in addition to selected marker genes. This gives a broader insight and is helpful for the field. This could easily be a supplemental figure.*

We thank the reviewer for this comment and completely agree that a broader cell enrichment list would be helpful. Although the reference suggested by the reviewer is an excellent one, we chose an alternate, more closely related reference to generate our broader gene list and heat map:

<https://www.ncbi.nlm.nih.gov/pmc/articles/PMC8639352/>

We chose this paper because it uses methods slightly more comparable to ours (specifically, RNA-seq from mRNA bound to translating ribosomes from brain tissue, which gives a very different transcript

profile than traditional bulk RNA-seq). We have now incorporated a supplemental figure (Supplementary Figure 1E) that represents this more comprehensive neuronal gene list, and again demonstrate neuronal population enrichment and non-neuronal cell depletion from our TRAP-seq data.

- 3. In the comparison of the simultaneous and separate TRAP the limited overlap of DEGs identified (Figure 2F) is a bit concerning. This could be a statistical power issue arising from the different sample numbers (2,3 for simultaneous and 3,3 for TRAP). Perhaps a plot comparing the fold change between the two methods would help address this question of whether the different in numbers are a reflection of genes that just pass or just fail to pass statistical criteria. It is also not clear what exact statistical tests and multiple testing corrections are used in the RNA-Seq analysis.*

We acknowledge this particular concern and address it as best as possible in our revised version of the manuscript. Indeed, we believe Figure 2D and 2E show what the reviewer requests. We plot gene length vs log fold change for each method separately and show high similarity between the two methods. The goal of this comparison of technical approaches was to make the point that these approaches, although not generating identical gene lists, do generate differential gene expression data that are highly comparable with regard to the predicted biological functions from sequencing results. Importantly, we found that GO Biological Process categories had substantial functional overlap regardless of separate or simultaneous technical approaches. It should also be appreciated that the separately isolated samples were from a single hemisphere, whereas the SIT was performed on combined L and R hemispheres, which we demonstrate in subsequent data that hemisphere matters with regard to the effect of ELE and actual gene list generated from TRAP-seq. Also, in order to address concerns regarding the different sample numbers and DEGs assessed from our two different methodological approaches, we performed a Spearman's correlation, which showed a strong linear relationship between the two sets of data. To address the second part of this reviewer comment, in Line 241, we now state the following: "The differentially expressed genes (DEGs) were analyzed using DESeq2 and Student's t-tests to determine statistically significant differences in RNA-seq and detect potential biologically relevant log-fold changes." This information, as well as our log fold change cutoffs, is now included in our Methods section (Lines 1013-1015).

- 4. An advantage of the simultaneous isolation protocol should be that there is stronger correlation between epigenomic and translomic profiles in the simultaneous vs separate protocols. Can the authors compare the cut & run vs expression relationships (Fig 2) in the two methods to address this?*

We appreciate this comment. We have consulted with bioinformatics experts here at the UCI Genomics High Throughput Facility, and to our knowledge there are no existing, current quantitative methods for this type of direct correlation of TRAP-seq and CUT&RUN-seq data (but approaches are currently being developed). At this point, the bioinformatic approach to perform this type of comparison is beyond the scope of our manuscript, even though this type of direct comparison would absolutely enhance the validity of the Emx1-NuTRAP / SIT approach.

5. *Figure 3A - As described above, more detail on the statistical approach is needed here. Is the Venn a comparison of independent t-tests performed on each hemisphere? With the small sample sizes (n=2-3/group) and thus the low power, presumably after multiple testing correction, it is easy to get genes that barely pass the critical threshold in one comparison but just fail to reach the p-value cutoff in the other. To really get at this central question of laterality it would seem that a 2-Way ANOVA design is needed and the interaction term of ELE and Hemisphere would provide an unambiguous determination of laterality. This will give the clearest indication of differences in DEGs by hemisphere.*

Thank you for this comment. Per the reviewer's suggestion we performed a 2-Way ANOVA, which demonstrated a clear effect of early life exercise on differential gene expression. These data are now presented in the text in Lines 286-291. This analysis also demonstrated an effect of hemisphere but to about half the degree (in terms of number of genes passing a p value significance threshold of 0.05, 1583 versus 808 genes). A likelihood ratio test was also performed with DESeq2's settings package, which showed a similar result.

With regard to the question about the Venn diagrams, their data represent DESeq2's default interpretation of a Wald test where each comparison is not exactly independent (so, this is not independent t-tests). This is based on the following article:

<https://genomebiology.biomedcentral.com/articles/10.1186/s13059-014-0550-8>

Each circle of the Venn represents the number of genes identified as significant by our cutoff descriptions in the Methods (Lines 1013-1015). when comparing sedentary to exercise in each hemisphere. Our goal with Figure 3 was to determine whether using one hemisphere over the other had significant functional difference rather than just strict statistical difference. The GO analysis presented demonstrates that functional categorization is preserved regardless of hemisphere.

6. *The multiple way Venn diagrams (Fig 5A & A') could benefit from an upset plot presentation (<https://r-graph-gallery.com/upset-plot.html>). This would aid the reader in understanding the overlaps between sets.*

Thank you very much for this recommendation. We have indeed generated an upset plot based on the reviewer's suggestion, and this is now included in Supplementary Figure 3. We appreciate this visualization of the data and agree that it enhances clarity of understanding overlaps between data sets.

7. *A weakness in the overall study design in that for the ELE analysis in Figure 6 the analysis switches back to whole tissue analysis from TRAP and INTACT of a specific cell type. This reduces the interoperability of the data with the other experiments.*

We acknowledge this fact that our study design uses WT mice rather than Emx1-NuTRAP mice for the memory consolidation portion of the project (Figure 6 data). We now discuss this caveat in the Discussion section of the manuscript and highlight the point that important conclusions, specifically testable mechanistic gene / transcription factor candidates, can still be derived from this data to generate hypotheses. Furthermore, our genes that are implicated from the memory consolidation experiment had to have been identified from the Emx1-NuTRAP sequencing data in order to derive information about their "priming" by ELE.

Reviewer #2 (Remarks to the Author):

Raus et al have studied how adolescent engagement in aerobic exercise translates to improved hippocampal-dependent memory, and propose that it involves improved transcriptional dynamics through histone modifications. To facilitate this study, they describe an innovative approach to isolating mRNA and chromatin from excitatory hippocampus neurons using Emx1-NuTRAP transgenic mice. This novel protocol is worth publishing (Nature Methods or Nature Protocols), but the biological interpretations regarding transcriptional responses during memory consolidation are weak.

We appreciate this reviewer's detailed comments below and provide responses.

Major points of concern:

1. *The authors have described the SIT protocol well. They have performed adequate controls. The absence of s100b in their initial cell sorting study significantly boosts the utility of this novel technique. One concern/query I have is regarding the concentration of TRAP-isolated mRNA yielded, which appears to be rather low <15ng/uL. Less specific methods of total RNA isolation typically yields >300ng/uL for bilateral whole hippocampus tissue. Where is all the excess RNA lost to? Is it expected that nuclear/ribosome-associated mRNA represents such a small fraction of the total RNA pool in neurons?*

We thank the reviewer for highlighting this very important point. In our "SIT" protocol, this includes a TRAP isolation, which will have lower yields than traditional RNA-isolation which will include both nuclear and Cytoplasmic mRNA. We are only getting translating mRNA and a small portion of ribosomal RNA. In the TRAP isolation, the GFP tag will only be associated with translating mRNA and thus we also do not get lncRNA, pre-mRNA, small and microRNAs, etc . We also must consider the expression level of the EGFP-L10a in hippocampal neurons expressing Emx1, and the strength of the Emx1 as a driver of this expression which may also have an impact here. However, the quality of our RNA was very high, and our quantity was sufficient for downstream applications in this study.

2. *The individual yields for (TRAP/INTACT) SIT indicate that 5 out of 8 samples yielded 15+ng/uL. The consistency and efficiency of the process is arguably not optimal. Is this reflective of issues with methodology or samples?*

Thank you very much for highlighting this point. Respectfully, we do not think that there is an issue with either our hippocampal samples or our methodology in this case. Bead pulldowns are known to have high variability, however, this variability should not interfere with the final downstream application of input amount and sequencing. Since performing the experiments presented in this manuscript, we have subsequently performed >40 isolations with high reproducibility (average concentration of 4.36ng/ul with a standard deviation of 3.61ng/ul using the SIT protocol in much smaller tissue samples, e.g. unilateral dorsal hippocampal tissue). Because this is what we typically see, we are assured that the consistency of our protocol is robust.

3. *Five out of twelve samples of TRAP only isolation yielded more than the reported average 7+ng/uL, there was one sample that yielded 23ng/uL. There is also a great degree of variance in this data set too, which likely reflects the sensitivity of the technique, but also raises concerns*

about the quality of this data set and the appropriateness of claiming that TRAP alone yielded ~half of the SIT protocol consistent with using one hemisphere vs bilateral samples. These efficiency issues are critical to new protocols since it would greatly affect the design and reproducibility of data for future studies relying on this method, if published.

We are incredibly appreciative of this reviewer's evaluation and critique of the data presented in Supplemental Tables, and we understand this excellent reviewer's point. Indeed, the separate isolation sample that yielded 23ng/ul was a clear outlier compared to the RNA yields of the other separate isolation samples (once excluding this outlier, the data are very tight). When this particular sample had its sequencing library built, it yielded the lowest final library concentration, thus suggesting the possibility that it was likely lower in concentration than what was measured. However, the final quality check (concentration and size profiling) performed on this sample was well within acceptable standards for subsequent sequencing. Therefore, we decided to retain this sample in our data analysis using appropriate data analysis parameters for determining whether data quality made this sample a significant outlier for further processing, which it did not.

- 4. No objective measure of memory consolidation is provided, so the authors cannot claim any benefit of exercise on memory capacity. If anything, the subsequent molecular responses only reflect response to a novel environment or experimenter handling.*

Thank you for this important comment. The study design presented here is informed by our previously published work demonstrating objective measures of long-term hippocampal memory following early life exercise (Ivy et al., Scientific Reports 2020). In this study we found that mice that underwent ELE during the 4th-6th postnatal weeks had both enabled memory performance on Object Location Memory, the same task presented in the current paper (mice had expression of long-term memory after a subthreshold / 3 min learning stimulus). We also found a significant increase in hippocampal CA1 long-term potentiation, the cellular correlate of hippocampal memory. We therefore applied this experimental design to the cohort of mice presented here and asked the question about differential gene expression during memory consolidation in animals that had ELE prior to exposure to a hippocampal learning event. Without sacrificing the mice during the hour after a learning stimulus, we are unable to evaluate differential gene expression during hippocampal memory consolidation. Furthermore, because we have the Emx1-NuTRAP data identifying genes "primed" by ELE which then change their expression during memory consolidation, we discover gene-epigenetic regulatory mechanisms that may be key in the effect of ELE on enabling hippocampal memory. We have re-worded to reiterate this point in the Results section.

- 5. The emphasis on "early life" provision of exercise is lost due to the absence of an aged exercise group. The claims to 'transcriptional programs by ELE that could be unique to exercise timing..' is rather pre-emptive.*

We agree with the reviewer's comment here and have thus added to the sentence: "... transcriptional programs by ELE that could be unique to exercise timing. To determine if gene expression programs identified here are in fact unique to exercise timing, an adult exercised cohort would need to be included for comparison." (Line 394)

6. *Figure 4 title 'ELE leads to... during adolescence'. The mice were killed at 42 days of age. Since behavioural testing and molecular analysis were performed on 6 week old tissue, this is reflective of young adult, not the adolescent brain.*

Although we appreciate that there is not complete agreement in the field regarding which ages constitute adolescence vs adulthood, for consistency, we consider the following reference when defining adolescence in mice: <https://pubmed.ncbi.nlm.nih.gov/26816516/>

In this manuscript adolescence is defined as the period between P22-P60. Our studies were performed on tissue from P42 mice, which we consider to be of mid-adolescent age based on their behavioral immaturity and the beginnings of entering sexual maturity at this age.

Minor points:

7. *It was confusing for the initial SIT RINs reported in lines 149-150 to be stated as >8, but then an average was later reported in line 200.*

Thank you for this comment, it has now been corrected and we report this data as all RINs as >8 for all samples (Lines 155-156 and Lines 206-208).

8. *It is a shame that only male mice were studied.*

We highly prioritize sex as a biological variable in our laboratory and unfortunately due to logistical limitations were unable to include both sexes in this particular study. Subsequent (and ongoing) experiments in our lab incorporate both sexes, and also consider phase of estrus in interpreting our behavior studies. Despite this limitation, for this initial Emx1-NuTRAP project, we have established in male mice the technical aspects as a foundation for growth of this study and its findings (although it could have been established in females as well). Our goal in part was to reduce the number of variables for the purposes of establishing Emx1-NuTRAP and SIT, which we have done here and are continuing to establish in both sexes.

9. *Were there one or two running wheels per cage? Was there competition for the running wheels, as this could contribute to stress, jostling for dominance to access the wheels?*

Thank you for this question. As now described in detail in our Methods, we have only one wheel per cage. Indeed, in subsequent experiments in our lab we have individually tracked pair housed mice and found significant differences in running amounts. We have also video recorded these mice and do not see increased fighting (unpublished data and Valientes et al, 2021). Therefore, we do not see any indications of stress resulting from competition over use of the wheel.

10. *It would be appropriate to report brightness of behaviour rooms in LUX, rather than simply reduced lighting. Line 734.*

This has now been changed in the text to reflect the actual LUX in our behavior room, which was measured as 8.5-11 (Line 764).

11. *I feel the subsection headings do not draw sufficient attention to the main study of sedentary vs exercise conditions.*

Thank you for this comment, we have now revised most of the subsection headings to emphasize the ELE effects and transcriptomic/epigenomic findings resulting from ELE.

12. The finding of transcriptional symmetry between L vs R hippocampus is an important confirmation. However the authors only compared L vs R for the ELE group, It would be appropriate to first demonstrate the presence of transcriptional symmetry in the sedentary controls. The reported transcriptional response to exercise is largely agreeable with the literature.

Thank you for this suggestion. In our study design, we evaluated differential gene expression as a result of ELE within each hemisphere (sedentary vs ELE). In doing so, our baseline group for comparison, i.e. the sedentary controls, were represented for each hemisphere. Therefore, the inherent baseline gene expression differences between hemispheres were controlled for. Additionally, we performed our Spearman's correlations comparing the L vs R hippocampus on the sedentary control condition only for our DEGs as well as histone modifications. Indeed, these data were highly correlated between hemispheres.

13. It is not clear how the up-regulation of expression can be concurrent with an active histone mark AND a repressive mark for 14 genes (Results Line 408). Is there evidence in the literature for this seemingly conflicting occurrence?

Thank you for bringing attention to this point. There are a number of recent studies that support the occurrence of bivalent chromatin marks, in which a typically active histone mark may be paired with a repressive mark to repress rapid transcription until the repressive mark is removed. A particularly good example is a study from the Schaefer lab, titled "Polycomb repressive complex 2 (PRC2) silences genes responsible for neurodegeneration": <https://pubmed.ncbi.nlm.nih.gov/27526204/>

In this study, non- medium spiny neuron (MSN) genes show a PRC2-dependent bivalent chromatin signature with both repressive and activating histone marks, whereas MSN-identity genes are marked monovalently. In neurodegenerative states, bivalent chromatin can become derepressed and leads to upregulation of non-MSN genes involved in cell death and senescence. This phenomenon and other findings give credence to our hypothesis that we may see a signature of (both mono- and bivalent) chromatin marks induced by early life exercise to promote gene expression that may facilitate neuronal function and memory. However, this has not been studied extensively in the context of experience-dependent changes in neuronal function and would be a future direction of our research.

REVIEWERS' COMMENTS:

Reviewer #1 (Remarks to the Author):

The authors have generally addressed the critiques (detailed below). Interesting aspects of the work are the technical methods and question regarding potentially priming effects of early exercise. Principle limitations are small sample sizes which must negatively affect statistical power for some points and the lack of the new model being used in the ultimate experiment.

Previous critiques:

Reviewer #1

1. GFP+/Thy- cells – That these could represent immature neurons is certainly a possibility. That immature neurons represent 1/3 of all of the labeled neurons seems less likely but this is offered and readers can interpret from there.
2. Broader sets of marker genes – addressed
3. Commonalities of DEGs between simultaneous and separate TRAP approaches – The laterality issue may certainly underlie some of the limited concordance in DEGs. Figure 2D and 2E show similar distributions to the populations but not a direct comparison. The methods state that a Benjamini Hochberg correction is used.
4. Comparison of simultaneous and separate DEG and Cut & Run data – This remains less than fully convincing as things like the GO analysis is very general (most prevalent DEG class is 'cellular process'). This reviewer would disagree that in fact the relationships between RNA and Cut&Run can be generated for each data collection type and then compared – even qualitatively.
5. Addressed
6. Addressed
7. This will just be a limitation to the study that the ultimate payoff of the NuTRAP systems for this type of study will come later

Reviewer #2

- 1-3 RNA yield/purity addressed
 4. Memory consolidation addressed
 5. Limitation stated
 6. Clarification on terminology definitions
- Minor points: addressed